# TMEM87a/Elkin1, a component of a novel mechanoelectrical transduction pathway, modulates melanoma adhesion and migration

Amrutha Patkunarajah[1,2†], Jeffrey H Stear[1,3†], Mirko Moroni[3‡], Lioba Schroeter[1,2], Jedrzej Blaszkiewicz[3], Jacqueline LE Tearle[1], Charles D Cox[4,5], Carina Fürst[3], Oscar Sánchez-Carranza[3], María del Ángel Ocaña Fernández[3§], Raluca Fleischer[3], Murat Eravci[6], Christoph Weise[6], Boris Martinac[4,5], Maté Biro[1,7], Gary R Lewin[3], Kate Poole[1,2,3,7]*

[1]EMBL Australia Node in Single Molecule Science, School of Medical Sciences, University of New South Wales, Sydney, Australia; [2]Cellular and Systems Physiology, School of Medical Sciences, University of New South Wales, Sydney, Australia; [3]Max Delbrück Center for Molecular Medicine, Berlin-Buch, Germany; [4]Victor Chang Cardiac Research Institute, Sydney, Australia; [5]St Vincent's Clinical School, University of New South Wales, Darlinghurst, Australia; [6]Freie Universität Berlin, Institute of Chemistry and Biochemistry, Berlin, Germany; [7]ARC Centre of Excellence in Advanced Molecular Imaging, University of New South Wales, Sydney, Australia

*For correspondence:
k.poole@unsw.edu.au

†These authors contributed equally to this work

Present address: ‡Bayer AG, Wuppertal, Germany; §Institute of Physiology, University of Freiburg, Freiburg, Germany

Competing interests: The authors declare that no competing interests exist.

**Abstract** Mechanoelectrical transduction is a cellular signalling pathway where physical stimuli are converted into electro-chemical signals by mechanically activated ion channels. We describe here the presence of mechanically activated currents in melanoma cells that are dependent on TMEM87a, which we have renamed Elkin1. Heterologous expression of this protein in PIEZO1-deficient cells, that exhibit no baseline mechanosensitivity, is sufficient to reconstitute mechanically activated currents. Melanoma cells lacking functional Elkin1 exhibit defective mechanoelectrical transduction, decreased motility and increased dissociation from organotypic spheroids. By analysing cell adhesion properties, we demonstrate that Elkin1 deletion is associated with increased cell-substrate adhesion and decreased homotypic cell-cell adhesion strength. We therefore conclude that Elkin1 supports a PIEZO1-independent mechanoelectrical transduction pathway and modulates cellular adhesions and regulates melanoma cell migration and cell-cell interactions.

## Introduction

Cells sense and respond to their physical surroundings by converting mechanical inputs into biochemical signals, a process referred to as mechanotransduction. The most rapid mode of mechanotransduction is mediated by mechanically activated (MA) ion channels that are activated within milliseconds of a stimulus, resulting in a localised flow of ions across the plasma membrane, thus converting physical stimuli into electrical signals. The discovery of the mammalian MA channels, PIEZO1 and PIEZO2 (*Coste et al., 2012*; *Coste et al., 2010*), has highlighted the diverse array of cells and tissues that express such ionotropic force sensors (*Albuisson et al., 2013*; *Florez-Paz et al., 2016*; *Hung et al., 2016*; *Maksimovic et al., 2014*; *Martins et al., 2016*; *Miyamoto et al., 2014*; *Ranade et al., 2014*; *Rocio Servin-Vences et al., 2017*; *Yang et al., 2016*).

**eLife digest** When cells receive signals about their surrounding environment, this initiates a chain of signals which generate a response. Some of these signalling pathways allow cells to sense physical and mechanical forces via a process called mechanotransduction. There are different types of mechanotransduction. In one pathway, mechanical forces open up specialized channels on the cell surface which allow charged particles to move across the membrane and create an electrical current.

Mechanoelectrical transduction plays an important role in the spread of cancer: as cancer cells move away from a tumour they use these signalling pathways to find their way between cells and move into other parts of the body. Understanding these pathways could reveal ways to stop cancer from spreading, making it easier to treat. However, it remains unclear which molecules regulate mechanoelectrical transduction in cancer cells.

Now, Patkunarajah, Stear et al. have studied whether mechanoelectrical transduction is involved in the migration of skin cancer cells. To study mechanoelectrical transduction, a fine mechanical input was applied to the skin cancer cells whilst measuring the flow of charged molecules moving across the membrane. This experiment revealed that a previously unknown protein named Elkin1 is required to convert mechanical forces into electrical currents. Deleting this newly found protein caused skin cancer cells to move more slowly and dissociate more easily from tumour-like clusters of cells.

These findings suggest that Elkin1 is part of a newly identified mechanotransduction pathway that allows cells to sense mechanical forces from their surrounding environment. More work is needed to determine what role Elkin1 plays in mechanoelectrical transduction and whether other proteins are also involved. This could lead to new approaches that prevent cancer cells from dissociating from tumours and spreading to other body parts.

The activation of these channels by externally applied mechanical stimuli underpins our senses of touch, proprioception and hearing (*Florez-Paz et al., 2016*; *Maksimovic et al., 2014*), and contributes to the physiological function of mechanoresponsive tissues such as the vasculature (*Evans et al., 2018*; *Li et al., 2014*), the urothelium (*Martins et al., 2016*; *Miyamoto et al., 2014*) and the cartilage (*Rocio Servin-Vences et al., 2017*). Recent evidence has suggested that PIEZO1 is not only activated by exogenously applied forces but can also be activated by cell-generated forces (*Blumenthal et al., 2014*; *Ellefsen et al., 2019*; *Nourse and Pathak, 2017*). Thus, MA channels may be involved in inside-out mechanical signalling and contribute to the ability of cells to probe the physical nature of their microenvironment. Because PIEZO channels do not account for all mammalian MA channel activity, identifying additional MA channels and characterising their function represent important outstanding challenges in the field (*Dubin et al., 2017*).

Inside-out mechanical signalling has been implicated in the development and metastasis of cancers. The stiffening of the local microenvironment (*Kai et al., 2016*), changes in cell packing and cell-cell adhesions (*Cichon et al., 2015*) and the tumour topology (*Lee et al., 2016a*) have all been shown to impact tumourigenicity or invasiveness. In the case of melanoma, cancer progression/transformation is directly linked to mechanical changes at a cellular level: cells become more compliant (*Jonas et al., 2011*), exhibit increased contractility (*Paszek et al., 2005*; *Sanz-Moreno et al., 2011*) and display altered morphology during transformation (*Poole and Müller, 2005*). During the process of metastasis, cells must break away from the primary tumour and navigate microenvironments with diverse physical properties including: confined pores in the ECM, the planar interface of the basement membrane, or tracks and channels formed from ECM fibres or the earlier passage of cancerous cells (*Wolf et al., 2009*). Given the mechanical changes during melanoma development and metastasis it is important to characterise mechanotransduction molecules in these cells, including MA ion channels.

We report the identification of MA ion channel activity at the cell-substrate interface in WM266-4 metastatic melanoma cells, using elastomeric pillar arrays to apply fine mechanical stimuli to cells via their connections to the subjacent matrix (*Poole et al., 2014*; *Sianati et al., 2019*), thus simulating deflections arising from cell-generated forces. Simultaneous recordings using whole-cell patch-clamp electrophysiology allowed us to directly measure the resulting MA currents. We have identified a

crucial molecule (TMEM87a) required for this channel activity and have renamed this polypeptide Elkin1, from the Greek word Elko, meaning 'to pull'. The deletion of *Elkin1* resulted in altered cell migration and increased interaction forces between melanoma cells and laminin 511 (LM511), a functionally important extracellular matrix (ECM) molecule. In addition, *Elkin1* deletion modulated cell-cell interactions, leading to facilitated dissociation of Elkin1-KO cells from organotypic spheroids.

## Results

### Measuring mechanically activated ion currents in melanoma cells

To establish whether melanoma cells exhibit MA channel activity, metastatic WM266-4 melanoma cells (originally isolated from a secondary tumour) were cultured on uncoated pillar arrays made of polydimethylsiloxane (PDMS). Mechanical stimuli were applied directly to cell-substrate contact

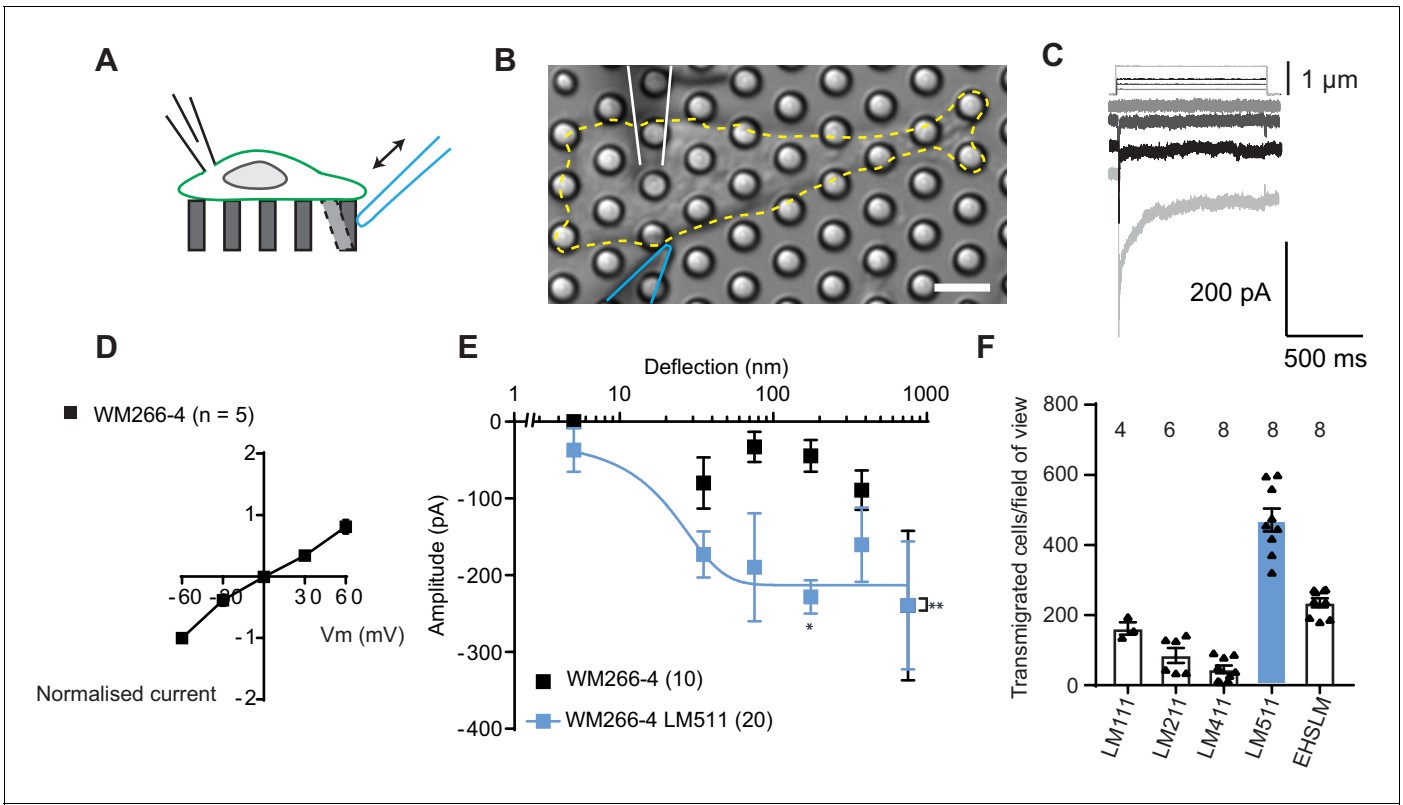

**Figure 1.** Measuring MA currents in WM266-4 melanoma cells. (A) Schematic of pillar array experiment. Cells were cultured on top of an array of elastomeric cylinders. Whole-cell patch-clamp was used to study the currents within the cell when stimuli were applied directly at the cell-substrate matrix by serially deflecting an individual pilus subjacent to the cell. (B) Bright-field image of a WM266-4 cell (outlined by dashed, yellow line) cultured on top of a pillar array. The microelectrode is outlined in white and the stimulating probe in blue. Scale bar = 10 μm. (C) Representative traces of inward MA currents activated in WM266-4 cells in response to increasing deflections. (D) Average current-voltage relationships of deflection-activated currents in WM266-4 cells (mean ± s.e.m., n = 5 cells). (E) Stimulus-response plots for WM266-4 cells on uncoated arrays (mean ± s.e.m., n = 10 cells) and WM266-4 cells on pillar arrays coated with LM511 (n = 20 cells). WM266-4 cells on LM511-coated arrays were more sensitive to pillar deflections than WM266-4 cells on uncoated arrays (ordinary two-way ANOVA, n = 20 and 10 cells respectively, **p=0.005; Sidak's multiple comparison, *p=0.02). (F) Transwell analysis of LM isoforms and their ability to promote transmigration. Note, LM511 supported the highest degree of transmigration, in comparison with other LM isoforms, LM111, LM211, LM411 and EHS-LM. See *Figure 1—figure supplement 1* for a comparison of mechanically evoked currents in WM115 versus WM266-4 cells and *Figure 1—figure supplement 2* for analysis of PIEZO1 contribution of mechanically evoked currents in WM266-4 cells.

The online version of this article includes the following source data and figure supplement(s) for figure 1:

**Source data 1.** Source data for details of current kinetics.
**Figure supplement 1.** MA currents in melanoma cell lines.
**Figure supplement 2.** Knockdown of *PIEZO1* in WM266-4 cells.

points by physically deflecting a single pilus subjacent to the cell (*Figure 1A,B*) and the electrical response of the cell was monitored using whole-cell patch-clamp. Deflection-activated currents were measured in all WM266-4 cells (10/10) and the current amplitude increased with increasing stimulus size (*Figure 1C*). Variable inactivation kinetics were measured (*Figure 1—figure supplement 1*, *Figure 1—source data 1*), as also previously demonstrated for PIEZO1-mediated currents activated by substrate deflection (*Poole et al., 2014*; *Sianati et al., 2019*). The reversal potential, as determined from a current-voltage relationship for the peak MA current, was +6.6 mV, indicating that the underlying current was passed by a non-selective cation channel (*Figure 1D*). Stimulus-response plots were generated by calculating the precise pillar deflection for each applied stimulus (*Figure 1E*). We additionally tested whether MA currents were activated in WM115 melanoma cells (isolated from the primary tumour from the same patient as WM266-4). Larger deflections were required to activate currents in WM115 cells, compared to WM266-4 (*Figure 1—figure supplement 1*). These data demonstrate that displacements at the interface between melanoma cells and their substrate evoke MA currents.

To examine whether MA channel activity is correlated to cell migration speeds, we performed pillar-array experiments using LM511, a substrate that supports the migration of metastatic melanoma cells (*Oikawa et al., 2011*). We first confirmed that LM511 promotes increased migration of WM266-4 cells using transwell assays, compared to other LM isoforms (*Figure 1F*). We then repeated the analysis of MA channel activity in WM266-4 cells cultured on pillar arrays coated with LM511 and noted that current kinetics were unchanged (*Figure 1—source data 1*). However, MA currents were more sensitive when evoked in WM266-4 cells attached to LM511 (*Figure 1E*). Under these conditions, current saturation occurred within the stimulus range, allowing us to use a Boltzmann sigmoidal fit to determine the MA current sensitivity. Half-maximal activation of MA currents was seen with approximately 18 nm of substrate deflection (Effective deflection ED50; standard error = 20.5 nm). These data indicate a correlation between migratory properties and the MA current sensitivity to deflections applied at cell-substrate contact points. The robust MA current activation observed in cells cultured on LM511 also provided an excellent system to investigate the molecules required for this mechanoelectrical transduction.

PIEZO1 is an obvious candidate for mediating this activity, in particular because the biophysical characteristics of the observed deflection-activated currents are consistent with those previously described for this channel (*Poole et al., 2014*). To directly test if PIEZO1 mediates the MA current in WM266-4 cells, we knocked down *PIEZO1* expression and examined whether these currents were still detectable. In *PIEZO1* knock-down cells cultured on LM511-coated pillar arrays, MA currents were activated in response to pillar deflection in 10/10 cells measured and the resulting stimulus-response curves were similar to controls (*Figure 1—figure supplement 2*). Similarly, treatment with Ruthenium Red (RR, a channel blocker that inhibits PIEZO1 and TRP channels) did not inhibit these MA currents (*Figure 1—figure supplement 2*). From these data we conclude that PIEZO1 is unlikely to be the MA channel responsible for the current measured in WM266-4 cells.

## Mechanically evoked currents in WM266-4 cells are dependent on Elkin1

To identify novel MA channels in WM266-4 cells, we undertook a proteomic-based strategy. Given that both intracellular (*Poole et al., 2014*; *Zhang et al., 2017*) and extracellular proteins (*Chiang et al., 2011*) can tune the sensitivity of MA channels, we analysed the proteome of WM266-4 cells rather than taking a comparative proteomic approach (*Supplementary file 1*). Two known non-selective cation channels were identified, PIEZO1 and TRPV2. However, RR is a channel blocker of both PIEZO1 (*Coste et al., 2012*) and TRPV2 (*Caterina et al., 1999*), indicating that neither likely mediates the deflection-evoked currents in WM266-4 cells (*Figure 1—figure supplement 2*). We then examined the proteomics data for proteins of unknown function with four or more predicted transmembrane (TM) domains. We prioritised the investigation of Elkin1 due to its expression in melanoma cells but not healthy melanocytes, its expression in additional mechanosensitive cells (Alveolar Type II cells) and its upregulation in additional human cancers (Human Protein Atlas [*Uhlén et al., 2005*] available from www.proteinatlas.org). We generated miRNA constructs targeting *Elkin1* and found that knockdown of *Elkin1* transcript resulted in a dramatic reduction in MA currents to deflections up to 1000 nm (*Figure 2A,B*). These data suggested that Elkin1 contributes to MA currents in melanoma cells.

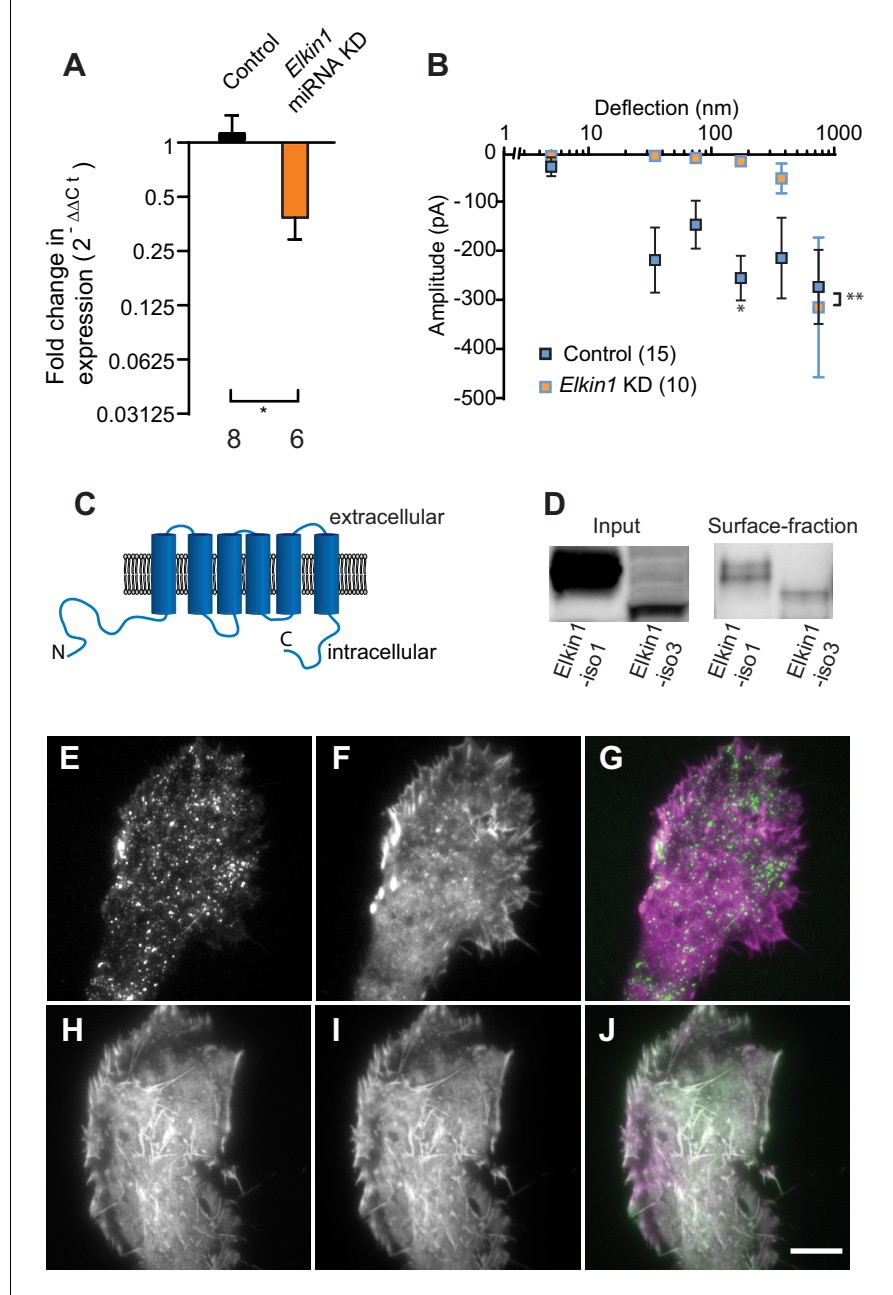

**Figure 2.** TMEM87a/Elkin1 in WM266-4 cells. (**A**) Transfection of WM266-4 cells with a plasmid encoding miRNA targeting *Elkin1* leads to a reduction in *Elkin1* transcript (Mann-Whitney, control n = 8, KD n = 6, *p=0.013). (**B**) Knockdown of *Elkin1* leads to a significant decrease in MA currents in WM266-4 cells, in comparison with controls (two-way ANOVA, control n = 15, *Elkin1* KD = 10, **p=0.004; Sidak's multiple comparison, *p=0.02) (data presented as mean ± s.e.m.). (**C**) Transmembrane topology prediction of *hs*Elkin1-iso1, with 6 TM domains (highest probability prediction of TM domains). (**D**) Western-blot analysis of samples prepared from HEK-293T cells overexpressing *hs*Elkin1-isoform1 or *hs*Elkin1-isoform3. The surface fraction was isolated by pull-down of biotinylated proteins after surface labelling. Note, both *hs*Elkin1-isoform1 and *hs*Elkin1-isoform3 are present at the cell surface. See *Figure 2—figure supplement 1* for full blot. TIRF images of (**E**) *hs*Elkin1-iso1-GFP, (**F**) Lifeact mCherry and (**G**) overlay in WM266-4 cells. Note that *hs*Elkin1-iso1 is present in foci as well as the plasma membrane. TIRF images of (**H**) *hs*Elkin1-iso3-GFP, (**I**) Lifeact mCherry and (**J**) overlay in WM266-4 cells. Note that *hs*Elkin1-iso3 is present in the plasma membrane and associated with actin structures. Scale bar 10 µm. See *Figure 2—videos 1* and *2* for corresponding live-cell imaging and *Figure 2—figure supplement 2* for laser scanning confocal imaging of *hs*Elkin1-iso1/*hs*Elkin1-iso3 with a Golgi-RFP marker.

*Figure 2 continued on next page*

*Figure 2 continued*

The online version of this article includes the following video and figure supplement(s) for figure 2:

**Figure supplement 1.** Cell-surface biotinlyation of *hs*Elkin1-GFP fusion proteins.

**Figure supplement 2.** Visualisation of *hs*Elkin1-iso1 and -iso3 with laser-scanning confocal microscopy.

**Figure 2—video 1.** *hs*Elkin1-iso1-GFP dynamics in WM266-4 cells, as visualised with TIRF microscopy.
https://elifesciences.org/articles/53308#fig2video1

**Figure 2—video 2.** *hs*Elkin1-iso3-GFP dynamics in WM266-4 cells, as visualised with TIRF microscopy.
https://elifesciences.org/articles/53308#fig2video2

Three human isoforms (representing splice variants) of Elkin1 have been identified: isoforms 1 and 3 (555 and 494 aa respectively), contain six predicted TM domains (*Figure 2C*). Isoform 2 (181 aa) does not contain any predicted TM domains and was not examined in this study. We cloned *hs*Elkin1-iso1 and *hs*Elkin1-iso3 from WM266-4 cDNA and generated C-terminal GFP fusion constructs. We confirmed the plasma membrane localisation of these two isoforms in transiently transfected HEK-293T cells using cell-surface biotinylation followed by Western blot analysis (*Figure 2D*, *Figure 2—figure supplement 1*). Both isoforms were present at the plasma membrane, as well as the intracellular fraction. To further examine the subcellular distribution of *hs*Elkin1, we transiently expressed these plasmids in WM266-4 cells and imaged the cells using total internal reflection fluorescence (TIRF) microscopy, which limits fluorescence excitation to molecules near the cell-substrate interface. *hs*Elkin1-iso1-GFP localised to both the membrane and a dispersed population of dynamic foci, while *hs*Elkin1-iso3-GFP assembled into structures that were co-labelled by Lifeact-mCherry. (*Figure 2E–J*; *Figure 2—videos 1* and *2*). Laser-scanning confocal imaging to visualise *hs*Elkin1-iso1-GFP throughout the cell revealed that this protein also localised to the Golgi apparatus (as previously described [*Hirata et al., 2015*]): in contrast, *hs*Elkin1-iso3-GFP was enriched in actin-based ruffles that were present at the cell periphery (*Figure 2—figure supplement 2*). The presence of Elkin1 in the plasma membrane, as described here, is consistent with Elkin1 forming an integral component of a mechanoelectrical transduction pathway.

## Elkin1 activation in a heterologous system

To test the hypothesis that Elkin1 contributes to mechanoelectrical transduction, we examined whether its expression in a heterologous cell system is sufficient to reconstitute MA currents. Untagged Elkin1 isoforms were overexpressed in the HEK-293T P1KO cell line (*Lukacs et al., 2015*), which lacks functional PIEZO1. The control cells exhibited no current activation in response to stimuli within our deflection range of 1–1000 nm (0/8 cells). However, mechanically activated currents were detected in response to similar stimuli in HEK-293T P1KO cells expressing either *hs*Elkin1-iso1 or *hs*Elkin1-iso3 (8/8 and 9/9 respectively) (*Figure 3A,B*). The observed current kinetics were consistent with direct mechanical activation (*Figure 3—source data 1*) and the mechanically evoked currents measured in WM266-4 cells (*Figure 1—source data 1*). We confirmed that we could mechanically evoke currents in a second cell line lacking PIEZO1, N2a $^{Piezo1-/-}$ (*Moroni et al., 2018*), where 6/8 cells expressing *hs*Elkin1-iso1 responded to pillar deflection compared to 4/10 control cells (*Figure 3—figure supplement 1*).

To investigate whether cells expressing Elkin1 were sensitive to alternative modes of mechanical stimuli, we used cellular indentation and high-speed pressure-clamp (HSPC). The HEK-293T P1KO cell line is unresponsive to indentation, (0/6 cells responding) as previously described (*Dubin et al., 2017*; *Figure 3C,D*); in contrast, MA currents were evoked in cells expressing either human Elkin1 isoform (*hs*Elkin1-iso1 6/6 cells, *hs*Elkin1-iso3 5/5 cells responding) (*Figure 3C,D*). Indentation-evoked currents were only measured in a small fraction of N2a $^{Piezo1-/-}$ cells expressing *hs*Elkin1-ios1 or *hs*Elkin1-iso3 (1/10 and 1/12 responding respectively compared to 0/8 control cells, *Figure 3—figure supplement 1*). When membrane stretch was applied using high-speed pressure-clamp (HSPC), no currents were evoked in negative controls or HEK-293T cells overexpressing human Elkin1 (negative pressure: *hs*Elkin1-iso1 0/11, *hs*Elkin1-iso3 0/13, positive pressure: *hs*Elkin1-iso1 0/5, *hs*Elkin1-iso3 0/5 responding) (*Figure 3E*). In contrast, both positive and negative pressure evoked currents in all positive controls where PIEZO1 was expressed (*Figure 3—figure supplement 2*). From these data we conclude that the expression of Elkin1 is sufficient to confer MA channel activity

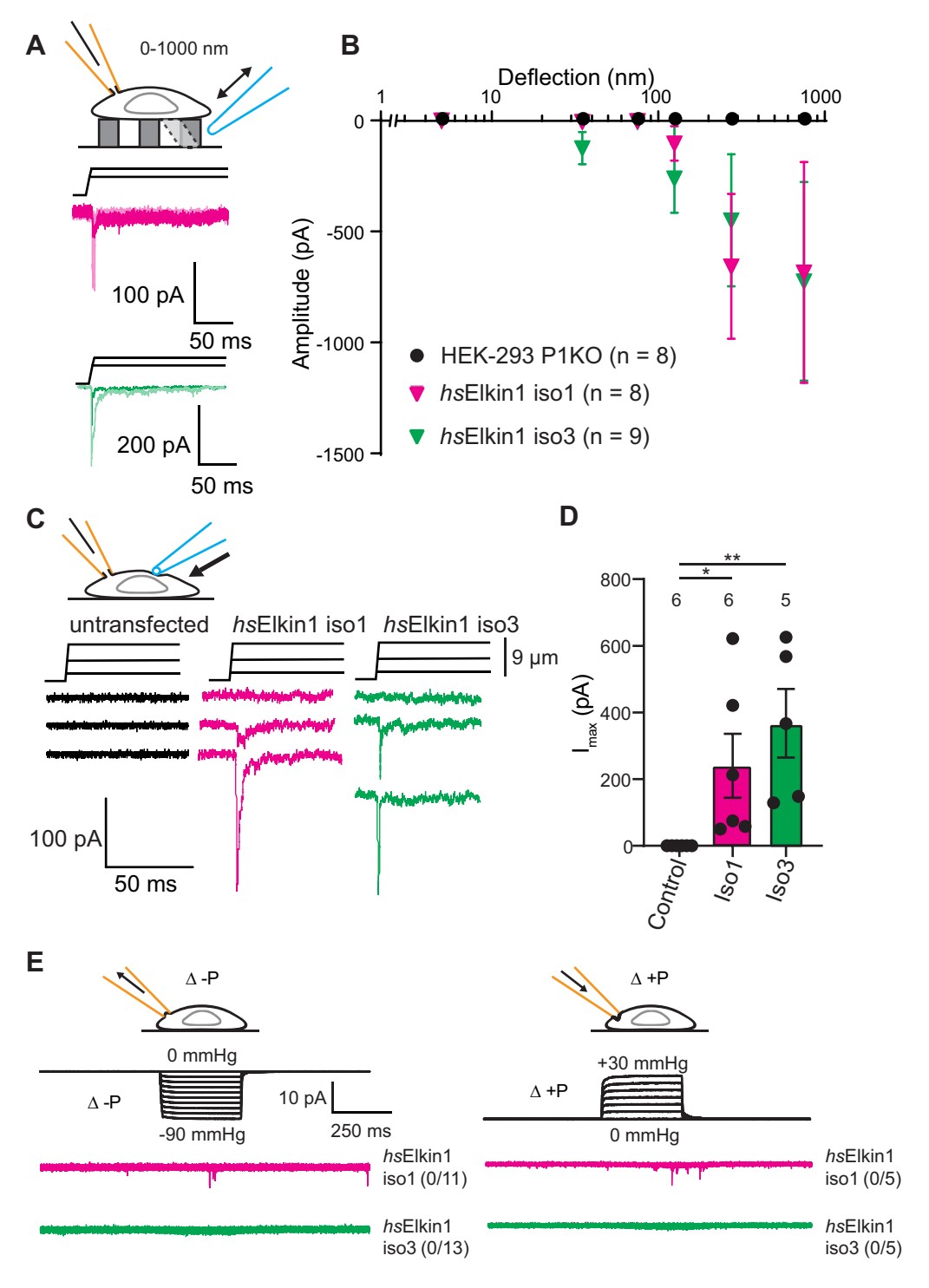

**Figure 3.** Elkin1-dependent currents can be activated in HEK-293T P1KO cells. (**A**) Example traces of Elkin1-dependent MA currents in HEK-293T P1KO cells (magenta: *hs*Elkin1-iso1; green: *hs*Elkin1-iso3). (**B**) Stimulus-response plots of HEK-293T P1KO cells (black circles, n = 8), HEK-293T P1KO cells expressing *hs*Elkin1-iso1 (magenta triangles, n = 8) or *hs*Elkin1-iso3 (green triangles, n = 9) (data are mean ± s.e.m.). Note that no currents were observed in the HEK-293T P1KO cells within the stimulus range of 1–1000 nm. (**C**) Example traces of indentation-activated MA currents in HEK-293T P1KO cells: control (black), *hs*Elkin1-iso1 (magenta), *hs*Elkin1-iso3 (green). Note that no MA currents are activated in response to indentation in HEK-293T P1KO cells in the absence of Elkin1. (**D**) Maximal current amplitude of indentation-activated currents was significantly larger in cells

*Figure 3 continued on next page*

*Figure 3 continued*

expressing *hs*Elkin1-isoform1 or *hs*Elkin1-isoform3 *versus* control cells (Student's *t*-test, control (n = 6) *versus* *hs*Elkin1-isoform1 (n = 6) *p=0.03, control *versus* *hs*Elkin1-isoform3 (n = 5) **p=0.003). Data are presented as mean ± s.e.m. with individual points overlaying bar graphs. (E) Example traces of cell-attached patch clamp recordings of Elkin-1 expressed in HEK-293T P1KO cells. Pressure stimuli ranging from 0 to -90 mmHg and 0 to +30 mmHg were applied using HSPC. Note that no stretch-activated currents were measured in any of the cells (negative pressure: control (black) = 0/8 cells, *hs*Elkin 1-iso1 (magenta) = 0/11 cells, *hs*Elkin1-iso3 (green) = 0/13 cells; positive pressure *hs*Elkin 1-iso1 (magenta) = 0/5 cells and *hs*Elkin1-iso3 (green) = 0/5 cells) Cartoons of stimuli adapted from *Rocio Servin-Vences et al. (2017)*. See *Figure 3—figure supplement 1* for analysis of Elkin1 activation in a second cell line (N2a *Piezo1-/-*), *Figure 3—figure supplement 2* for HSPC analysis of PIEZO1 expressed in HEK-293T P1KO cells.

The online version of this article includes the following source data and figure supplement(s) for figure 3:

**Source data 1.** Physiological properties of currents recorded in HEK-293 P1KO cells.
**Figure supplement 1.** Electrophysiological characterisation of *hs*Elkin1-iso1, *hs*Elkin1-iso3 and *mm*Elkin1.
**Figure supplement 2.** High speed pressure clamp recordings in cells expressing *hs*PIEZO1.

---

to substrate deflection and cell indentation stimuli in a heterologous system lacking endogenous PIEZO channels, but not to membrane stretch.

The *Mus musculus* homolog of Elkin1-iso1 shares 93% sequence identity at the protein level with the human variant. However, we did not observe robust MA currents in HEK-293T P1KO cells following expression of mouse *mm*Elkin1-iso1 (5/8 cells responding, *Figure 4A,B*), indicating a difference

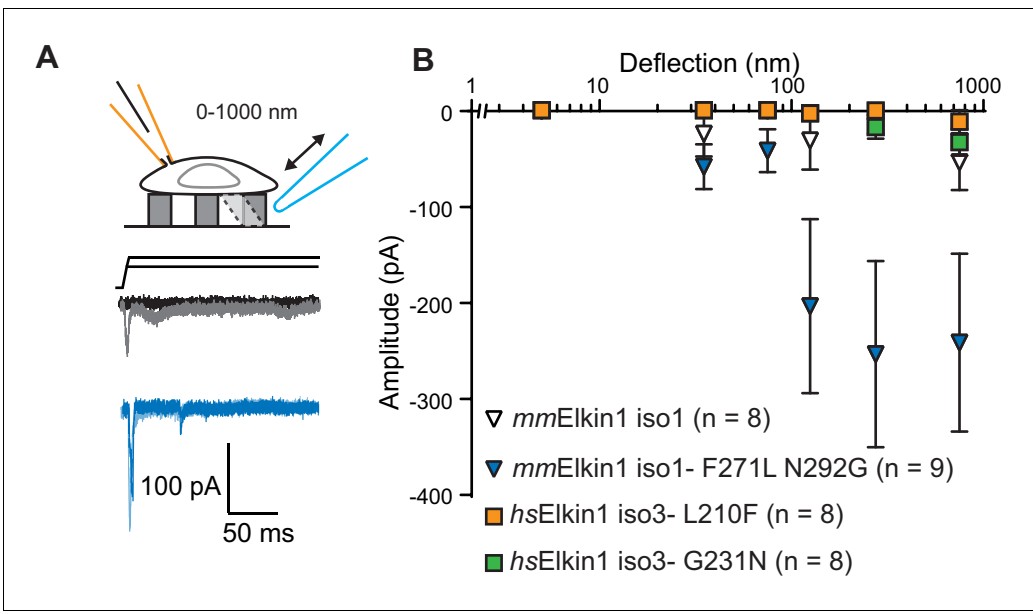

**Figure 4.** *hs*Elkin1 and *mm*Elkin1-dependent currents exhibit distinct mechano-sensitivity. (A) Example traces of *M. musculus* Elkin1-dependent MA currents in HEK-293T P1KO cells (grey: *mm*Elkin1-iso1; blue *mm*Elkin1-iso1 F271L N292G). (B) Stimulus-response plots of HEK-293T P1KO cells expressing: *mm*Elkin1-iso1 (open triangles, n = 8 cells), *mm*Elkin1-iso1 F271L N292G (blue triangles, n = 9 cells), *hs*Elkin1-iso3 L210F (orange squares, n = 8 cells), *hs*Elkin1-iso3 G231N, (green squares, n = 8 cells). Data points presented as mean ± s.e.m. Cartoons of stimuli adapted from *Rocio Servin-Vences et al. (2017)*. See *Figure 4—figure supplement 1* for sequence alignment of *hs*Elkin1 and *mm*Elkin1, *Figure 4—figure supplement 2* for surface biotinylation analysis of Elkin1 variants and *Figure 4—source data 1* for details on current kinetics.

The online version of this article includes the following source data and figure supplement(s) for figure 4:

**Source data 1.** Source data for details of current kinetics.
**Figure supplement 1.** Sequence alignment of human and mouse Elkin1 protein and the effect of N-terminal deletions on *hs*Elkin1 function.
**Figure supplement 2.** Cell-surface biotinlyation of Elkin1-GFP fusion proteins.

in sensitivity between these homologous proteins. These data were confirmed in the N2a $^{Piezo1-/-}$ background (*Figure 3—figure supplement 1*), with 11/26 cells responding. Alignments between the protein sequences (*Corpet, 1988*) revealed that the divergence in amino acid sequences primarily resides in the N-terminus of the proteins (*Figure 4—figure supplement 1*). However, expression of N-terminal truncation mutants of *hs*Elkin1 conferred similar MA currents to those of the wild type (WT) protein (*Figure 4—figure supplement 1*), suggesting that this region is not required for MA currents. To further characterise the differences between the mouse and human proteins, we focused on two non-conservative changes within the region containing the six predicted TM domains. Introducing the two human residues into the mouse polypeptide (*mm*Elkin1-iso1-F271L-N292G), led to robust MA channel activity, in contrast with WT *mm*Elkin1-iso1. Conversely, introducing either of the mouse residues into the human polypeptide (*hs*Elkin1-iso3-L210F or *hs*Elkin1-iso3-G231N) led to a reduction in the amplitude of the MA currents measured (*Figure 4A,B*), despite the fact that these variants were still present in the plasma membrane (*Figure 4—figure supplement 2*). These data indicate that two residues can account for the differences in activity between the mouse and human proteins, and that these two residues are required for the activation of robust Elkin1-dependent currents by substrate deflection.

## Elkin1 regulates cell migration

Having demonstrated that human Elkin1 is involved in mechanoelectrical transduction in melanoma cells, we sought to investigate whether the presence of this protein influences cell migration. To address this question, we used CRISPR/Cas9 to gene-edit cells such that no functional Elkin1-iso1/3 were expressed (*Figure 5—figure supplement 1*). Duplicate clonal populations of wild-type (WT) and Elkin1-knockout (KO) cells were isolated. Using pillar arrays, we observed reduced deflection-activated currents in the Elkin1-KO clones compared to the WT (*Figure 5A,B*). A residual current at large deflections was still present in some KO cells; we hypothesise that this effect is due to minor compensation from PIEZO1 (*Supplementary file 1* and as noted previously [*Rocio Servin-Vences et al., 2017*]) or the activity of an as-yet-identified MA channel. Nevertheless, these data confirm that Elkin1 is required for sensitive MA channel activity in WM266-4 cells.

We first tested whether Elkin1-dependent mechanotransduction regulates cell migration using a transwell assay. While transmigration of the WT clones was not different from control populations, both Elkin1-KO clones exhibited reduced transmigration (*Figure 5C*). This phenotype could be rescued by the transient overexpression of wild-type *hs*Elkin1-iso3. In contrast, overexpression of *hs*Elkin1-iso3 L210F (*Figure 5D*), a variant associated with significantly reduced MA currents (*Figure 4B*), did not rescue the transmigration defect. We additionally generated an Elkin1-KO clone from the A375 melanoma cell line (which has previously been shown to express functional PIEZO1 [*Hung et al., 2016*]). The A375 Elkin1-KO cells exhibited reduced transmigration, compared to the A375 WT cells (*Figure 5E*), indicating that the link between Elkin1 activity and migration is not restricted to WM266-4 cells and can be measured in a cell line where PIEZO1 is functionally active. We therefore conclude that disruption of Elkin1 not only impairs mechanosensitivity, but also cell migration.

## Elkin1 regulates unconfined (2D) and confined (*quasi*-1D) cell migration

We expanded our analysis to investigate how Elkin1 regulates different modes of migration. To examine how Elkin1 mediates the unconfined migration of individual cells on a 2D surface, we plated cells on dishes coated with LM511, and observed their movement using phase-contrast and epi-fluorescent microscopy. A representation of the migration tracks shows that the WT clones migrated further over the course of the experiment (*Figure 6A*). To test whether this effect was exclusive to migration on LM511-coated substrates, we repeated the experiments on dishes coated with poly-L-lysine (PLL) (*Figure 6B*). The Elkin1-KO clones exhibited reduced track mean speed on both substrates (*Figure 6C*) and a reduced distance migrated from origin (calculated as the Euclidean distance, *Figure 6D*).The percent reduction in the mean of the measured track mean speed was similar for experiments conducted on LM511 (Elkin1-KO cells 29% reduction in the mean) to PLL (Elkin1-KO cells 32% reduction). These data demonstrate that Elkin1 modulates cell migration in unconfined environments. To investigate the impact of Elkin1 on confined migration, stripes of LM511 were printed onto glass coverslips and the unprinted regions were passivated to block cell adhesion, thus

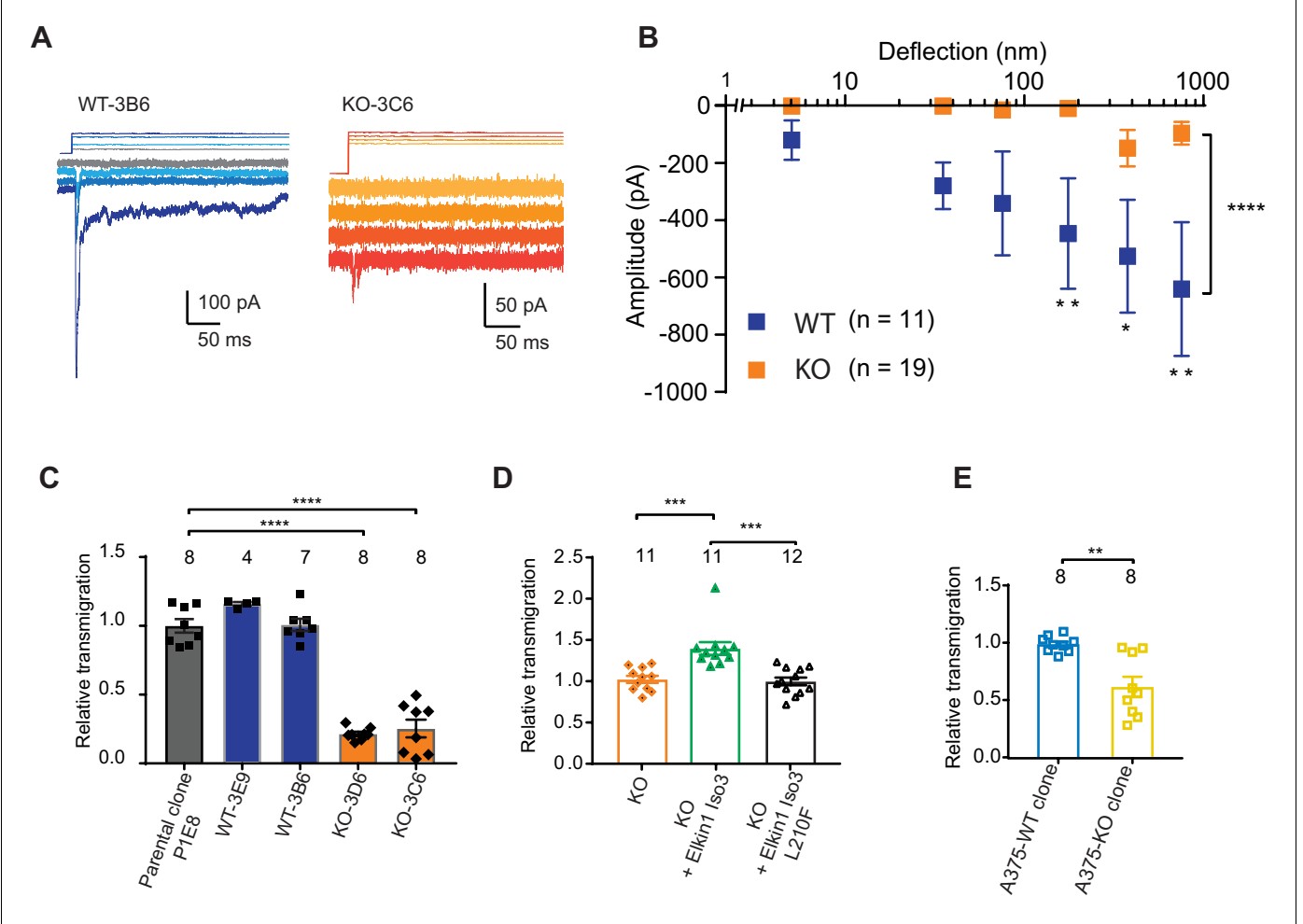

**Figure 5.** CRISPR/Cas9 deletion of Elkin1 inhibits MA currents and migration of WM266-4 cells. (**A**) Example traces of MA currents in WT and Elkin1-KO clones of WM266-4 cells on LM511-coated pillar arrays. A residual current is present in Elkin1-KO clones at large deflections. (**B**) Stimulus-response plots showing that MA currents are significantly inhibited in the Elkin1-KO clones in comparison with the WT clones (ordinary two-way ANOVA, Elkin1 KO = 19 cells, WT = 11 cells, ****$p<0.0001$; Sidak's multiple comparison, **$p=0.007$, *$p=0.03$, **$p=0.004$). (Data displayed as mean ± s.e.m.). (**C**) A transwell analysis of migration onto LM511-coated membranes shows that WT clones (3B6 and 3E9) were indistinguishable from WT controls, whereas Elkin1-KO clones (3C6 and 3D6) exhibited significantly reduced transmigration (one-way ANOVA, parental control n = 8 wells, 3E9 = 4 wells, 3B6 = 7 wells, 3D6 = 8 wells, 3C6 = 8 wells, ****$p<0.0001$; Dunnett's multiple comparisons, control vs 3C6, ****$p<0.0001$; control vs 3D6, ****$p<0.0001$, samples normalised against WT control). (**D**) The transmigration phenotype in the KO was rescued by overexpression of *hs*Elkin1-iso3, but not *hs*Elkin1-iso3-L210F (one-way ANOVA, KO control n = 11 wells, +*hs*Elkin1-iso3 = 12 wells, +*hs*Elkin1-iso3-L210F = 11 wells, ****$p<0.0001$; Dunn's multiple comparisons, Control vs +*hs*Elkin1-iso3, ***$p=0.0006$; +*hs*Elkin1-iso3 vs +*hs*Elkin1-iso3-L210F, ***$p=0.0002$, samples normalised against KO control). (**E**) In the A375 melanoma cell line, an Elkin1-KO clone also exhibited a significant decrease in transmigration onto LM511, in comparison with a WT control (unpaired *t*-test with Welch's correction, WT and Elkin1-KO = 8 wells, **$p=0.002$, samples normalised against WT control). (**C–E**) Individual data points overlay mean ± s.e.m. See *Figure 5—figure supplement 1* for CRISPR/Cas9 strategy and knockout clone validation.

The online version of this article includes the following figure supplement(s) for figure 5:

**Figure supplement 1.** CRISPR/Cas9 editing of WM266-4 cells.

restricting attachment to the printed regions (maximum width of 5 μm) (*Figure 6E*). During the experiment cells would occasionally detach and then reattach; our analysis only included periods when cells were attached and elongated on the patterned region. The mean migration speed was lower in the Elkin1-KO clones compared to WT (*Figure 6F*) (22% reduction in the mean Elkin1-KO track mean speed), indicating that disrupting Elkin1 inhibits the ability of cells to undertake both unconfined and confined migration on hard surfaces.

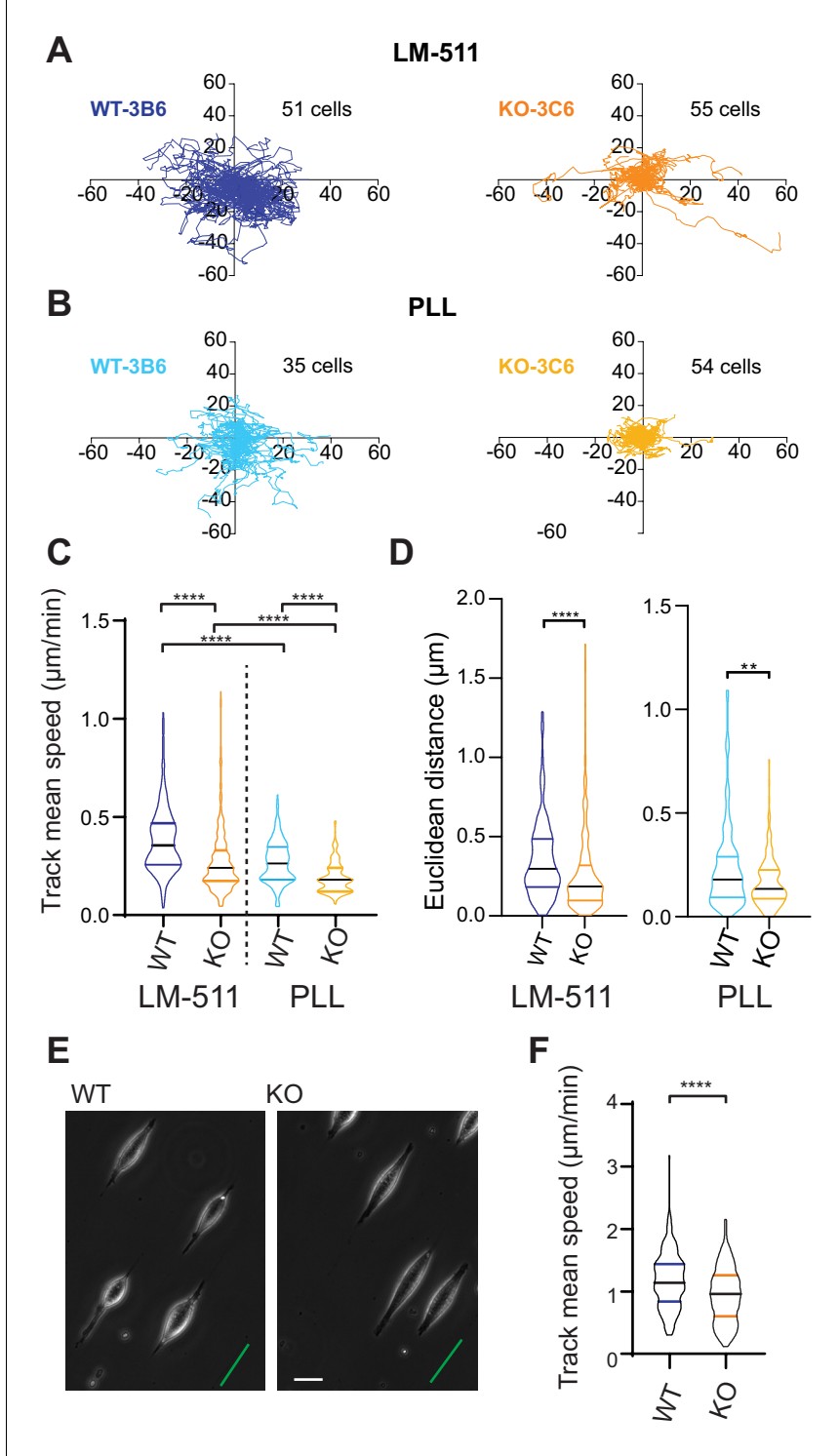

**Figure 6.** Elkin1-KO cells exhibit decreased migration on 2D and *quasi*-1D substrates. (**A**) Example tracks of cell movement on LM511 substrates of representative WT (3B6, 51 cells) and Elkin1-KO clones (3C6, 55 cells). (**B**) Example tracks of cell movement PLL substrates of representative WT and Elkin1-KO clones (3B6 = 35 cells, 3C6 = 54 cells). (**C**) The Elkin1-KO clones exhibited a significant decrease in mean track speed in comparison with WT clones on LM511 (Mann-Whitney, WT = 246 tracks, Elkin1-KO = 340 tracks, ****p<0.0001) and on PLL (Mann-Whitney, WT = 240 tracks, Elkin1-KO = 241 tracks, ****p<0.0001). In addition, WT clones exhibited a significantly higher mean track speed on LM511 compared with PLL (Mann-Whitney, LM511 n = 246 tracks, PLL = 240 tracks, ****p<0.0001) and the Elkin1-KO clones exhibited a significantly higher mean track speed on LM511 compared

*Figure 6 continued on next page*

*Figure 6 continued*

with PLL (Mann-Whitney, LM511 n = 340 tracks, PLL = 241 tracks, ****p<0.0001). See *Figure 6—figure supplement 1* for supporting experiments conducted using GFP-labelled cells. (D) The Euclidean distance calculated for the Elkin1-KO clones was significantly lower than WT clones on LM511 (Mann-Whitney, WT = 246 tracks, Elkin1-KO = 340 tracks, ****p<0.0001) and PLL globally coated substrates (Mann-Whitney, WT = 240 tracks, Elkin1-KO = 241 tracks, **p=0.0020). (E) Representative images of WT and Elkin1-KO clones attached to *quasi*-1D LM511 substrates, green line indicates direction of printed stripes, scale bar = 20 µm. (F) Elkin1-KO clones exhibited a significantly decreased mean track speed, in comparison with the WT clones (Mann-Whitney test, WT = 260 tracks, Elkin1-KO = 275 tracks, ****p<0.0001). (C,D,F) Data displayed as violin plots to represent relative distribution of data, black lines indicate median and coloured lines indicate quartiles. See *Figure 6—source data 1* for further details.

The online version of this article includes the following source data and figure supplement(s) for figure 6:

**Source data 1.** Source data for migration speeds and distances.
**Figure supplement 1.** GFP-expressing Elkin1-KO cells exhibit decreased migration on 2D substrates, in comparison with WT cells.

## Loss of Elkin1 facilitates dissociation of WM266-4 cells from organotypic spheroids

Given that cell dissociation from the primary tumour is required for tumour metastasis, we investigated the effect of Elkin1 deletion using *in vitro* organotypic spheroids. We selected representative WT and Elkin1-KO clones and labelled these cells using a viral vector expressing GFP. Spheroids formed from GFP-expressing cells were implanted in 3D collagen gels and imaged at 24, 48 and 72 hr post-implantation. At every time point, more Elkin1-KO cells had dissociated from the spheroid compared to WT (*Figure 7A–G*) and at 48 and 72 hr the Elkin1-KO cells were further away from the edge of the spheroid (*Figure 7H*, *Figure 7—figure supplement 1*). Imaging the first 12 hr after spheroid implantation highlighted the fact that Elkin1-KO cells dissociated more readily from the spheroid mass (*Figure 7—videos 1* and *2*). We investigated whether isolated Elkin1-KO cells embedded in 3D collagen gels exhibited a change in migratory properties to determine if an increase in migration speed could account for these data. In this 3D environment, the mean track speed was not different between WT and Elkin1-KO clones, however the track straightness was reduced in the Elkin1-KO clones (*Figure 7—figure supplement 2*). We additionally noted that the Elkin1-KO cells were less spherical as they broke away from the spheroid (*Figure 7I*, *Figure 7—figure supplement 1*) as they were in the 3D collagen gels (*Figure 7—figure supplement 2*)). However, these effects on sphericity and track straightness were moderate, with much of the data set overlapping. These data thus suggest that the increased distance of the Elkin1-KO cells from the spheroid is due to facilitated dissociation, not due to increased migration speed.

## Deletion of Elkin1 modulates cell binding forces

One model to account for the increased dissociation of cells from Elkin-1-KO spheroids as well as their altered migration properties is that deletion of Elkin1 changes physical cellular interactions. To test this idea we used atomic force microscopy (AFM) to measure unbinding forces after short-term contact between cells and LM511 or homotypic cell-cell contacts (*Hofschröer et al., 2017*; *Puech et al., 2006*). The unbinding of LM511 from Elkin1-KO cells required a larger force than the WT clone (37% increase in the mean of the maximum unbinding force) (*Figure 8A,B*). In contrast, lower unbinding force was measured after homotypic contact between Elkin1-KO cells, in comparison with WT cells (27% decrease in the mean of the maximum unbinding force) (*Figure 8C,D*). To determine if deletion of Elkin1 influenced cell-cell organisation over longer time scales, we created chimeric spheroids between Elkin1-KO and WT cells by mixing equal numbers of a GFP-labelled and an unlabelled clone. Regardless of which clone expressed the GFP, the spheroid was organised such that Elkin1-KO cells were found in the outer layer of the spheroid and the WT cells in the core (triplicate experiments, 23 spheroids total) (*Figure 8E,F*). To test whether the channel activity of Elkin1 is linked to cell partitioning within spheroids, we stably transfected Elkin1-KO cells with either *hs*Elkin1-iso3 or *hs*Elkin1-iso3-L210F. In chimeric spheroids the partitioning of cells was partially rescued in cells expressing *hs*Elkin1-iso3 but not *hs*Elkin1-iso3-L210F, where cells were evenly distributed throughout the resulting spheroids (duplicate experiments, 6 and 9 spheroids,

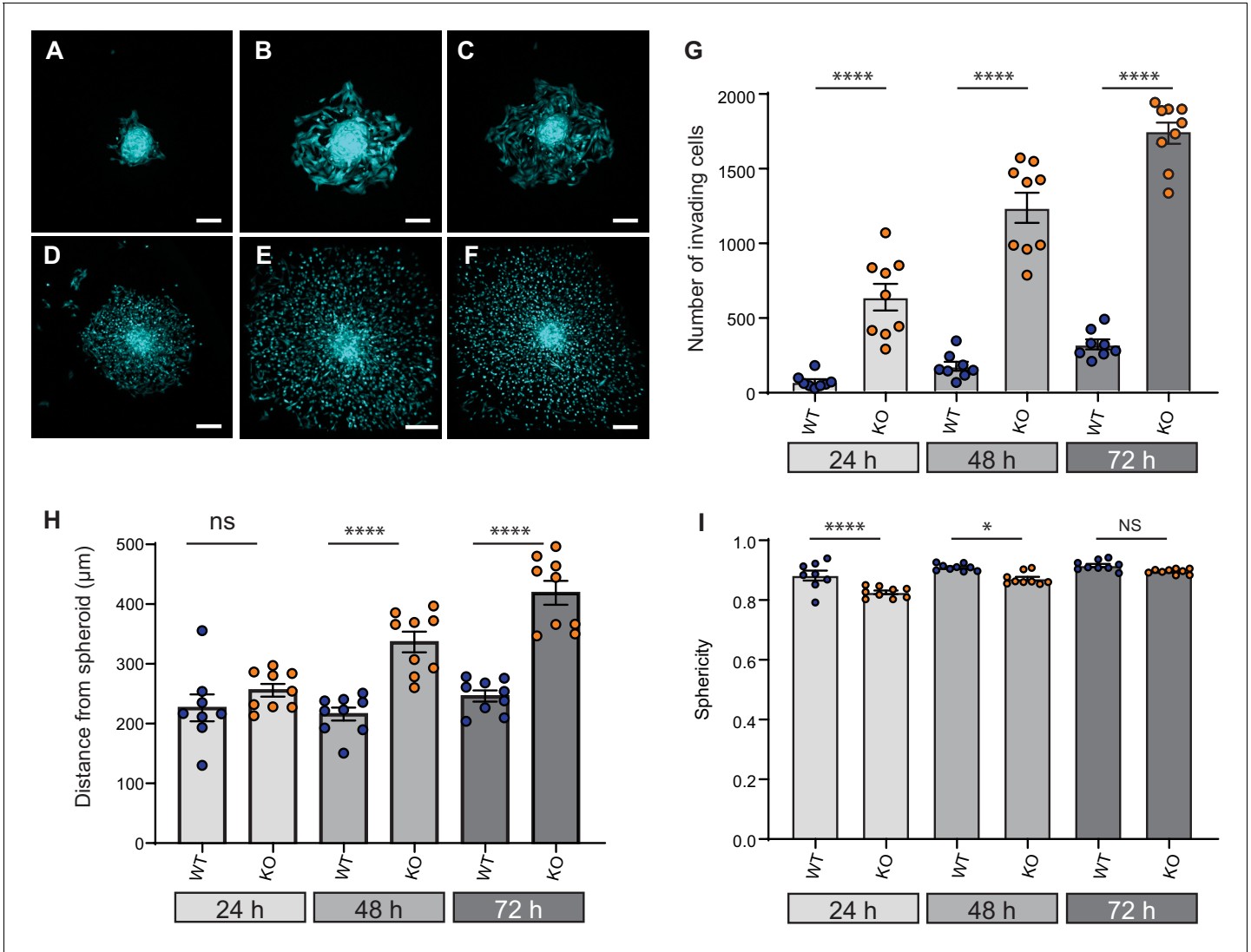

**Figure 7.** Elkin1-KO increases cell dissociation from organotypic spheroids. (**A–C**) Representative images of WT spheroids at 24 (**A**), 48 (**B**) and 72 (**C**) h post-implantation in 3D collagen I gel. See *Figure 7—videos 1* and *2* for live imaging of 0–12 hr post-implantation. (**D–F**) Representative images of Elkin1-KO spheroids at 24 (**D**), 48 (**E**), and 72 hr (**F**). Scale bars = 200 µm. (**G**) At all time points the number of Elkin1-KO cells that had dissociated from the spheroid was higher than for the WT cells. Data is presented as the mean ± s.e.m. with individual points representing the average number of invading cells for each spheroid (one-way ANOVA with Tukey's multiple comparison, WT = 9, Elkin1-KO = 9, 24, 48, 72 hr ****p=0.0001). (**H**) Average distance of cells from the edge of the spheroid. Data are presented as mean ± s.e.m. of distance from spheroid with overlay of points representing the average for each individual experiment The average distance per spheroid was significantly different at 48 and 72 hr, but not 24 hr (one-way ANOVA with Tukey's multiple comparison: WT = 9 spheroids, Elkin1-KO = 9 spheroids, 24 h p=0.78; 48 hr, ****p<0.0001; 72 hr, ****p<0.0001). See *Figure 7—figure supplement 1A* for data representing all individual cells. (**I**) Sphericity of cells that had invaded the collagen gel. Data are average sphericity of all cells within the collagen gel at each time point, presented as bar graphs with mean ± s.e.m. with an overlay of average for each spheroid measured. The WT cells were significantly more spherical than the Elkin1-KO clones at 24 and 48 hr (one-way ANOVA with Tukey's multiple comparison, WT = 9 spheroids, Elkin1-KO = 9 spheroids, 24 hr, ****p<0.0001; 48 hr, *p=0.012; 72 hr, NS, p=0.46). See *Figure 7—figure supplement 1B* for data representing all individual cells, *Figure 7—figure supplement 2* for migration data corresponding to isolated cells in 3D collagen gels and *Figure 7—source data 1* for further details.

The online version of this article includes the following video, source data, and figure supplement(s) for figure 7:

**Source data 1.** Source data for 3D migration properties.
**Figure supplement 1.** Data from all individual cells invading collagen gels from organotypic spheroids.
**Figure supplement 2.** The effect of Elkin1 deletion on migration in 3D collagen gels.
**Figure 7—video 1.** Dissociation of WM266-4 WT cells in organotypic spheroid assay.
https://elifesciences.org/articles/53308#fig7video1
**Figure 7—video 2.** Dissociation of WM266-4 Elkin1-KO cells in organotypic spheroid assay.

*Figure 7 continued on next page*

respectively) (*Figure 8—figure supplement 1*). Taken together these data indicate that deletion of *Elkin1* has a differential effect on cell-substrate *versus* cell-cell binding and regulates cell-cell associations in organotypic spheroids.

## Discussion

We have identified a novel mechanoelectrical transduction pathway in melanoma cells, activated by mechanical stimuli applied at the cell-substrate interface. The sensitivity of MA currents found in melanoma cells grown on LM511 was comparable to those found in the sensitive mechanoreceptors of the dorsal root ganglia required for fine touch (*Poole et al., 2014*). Furthermore, these molecular-scale pillar movements lie within the range of matrix displacements that arise due to cells pulling on their surroundings (*Legant et al., 2013*; *Legant et al., 2010*). In WM266-4 melanoma cells these deflection-activated currents were dependent on the Elkin1 protein. Elkin1 contains a LUSTR (Lung Seven Transmembrane) domain, which defines a family of proteins of as yet unknown function found in plants and animals, but not bacteria, archaea or viruses (*Edgar, 2007*). This LUSTR family also includes TMEM87b, a protein that shares 48% homology with *hs*Elkin1-isoform1. There is limited information regarding the physiological function of Elkin1, however the gene is expressed in diverse human tissues including organs where mechanical feedback is important, such as the lungs and the bladder (GTEx portal). Elkin1 was previously suggested to be a Golgi-associated protein that, when overexpressed, partially rescued a VPS54-KO endosome to trans-Golgi network (TGN) retrograde transport phenotype (*Hirata et al., 2015*). In contrast, we found that a fraction of both *hs*Elkin1-iso1 and -iso3 is localised to the plasma membrane, with *hs*Elkin1-iso1 also present within the Golgi. Elkin1 has also been demonstrated to be amongst the fraction of cell surface proteins that undergo N-linked glycosylation (*Park et al., 2018*), further supporting our data. Given that *Elkin1* knockdown did not inhibit endosome to TGN retrograde signalling (*Hirata et al., 2015*), but did ablate MA channel activity (shown here), we propose that Elkin1 is an essential component of a novel mechanoelectrical transduction pathway.

In support of this model, heterologous expression of Elkin1 was associated with the appearance of MA currents activated by substrate deflections in two cell lines in which *Piezo1* was deleted. The latency between stimulus and response (<2 ms) and the activation time constant (<1 ms) were similar to those reported for PIEZO1-mediated currents (*Poole et al., 2014*; *Rocio Servin-Vences et al., 2017*) and sufficiently rapid to suggest that the Elkin1-dependent current is directly activated by the mechanical stimulus (*Christensen and Corey, 2007*). In addition, none of the other MA non-selective cation channels (PIEZO2, TRPV4) were detected in WM266-4 cells, suggesting that Elkin1 either mediates these MA currents or modulates an unidentified MA channel. These data definitively demonstrate that Elkin1 is required for a novel, PIEZO1-independent mechanoelectrical transduction pathway. Mutation of a single residue in a predicted TM helix and of a single residue in a predicted intracellular loop led to a marked reduction in MA channel activity, without a concomitant reduction in surface localisation of the protein. Taken together these data indicate that Elkin1 exhibits several hallmarks of a novel MA channel (*Christensen and Corey, 2007*). However, it is possible that Elkin1 is an accessory molecule that modulates the activation of an, as yet uncharacterised, MA channel or an ion channel that requires additional proteinaceous tethers in order to be activated by mechanical inputs. Future studies with purified protein will be required to definitively test the hypothesis that Elkin1 is an ion channel activated by mechanical inputs.

In HEK-293T P1KO cells, human Elkin1 isoform1 and 3 were associated with a current activated by substrate deflection. Expression of mouse Elkin1 in the HEK293T P1KO cells was also associated with the *de novo* appearance of deflection activated currents, though these currents were of smaller amplitude. Channel activity could also be evoked by indentation (however, maximal currents were smaller than reported for the PIEZOs [*Coste et al., 2010*]) but not by membrane stretch applied using HSPC. This mechanical response profile is distinct to other MA ion channels: PIEZO1 and PIEZO2 respond to membrane stretch, cell indentation and substrate deflection (*Coste et al., 2010*;

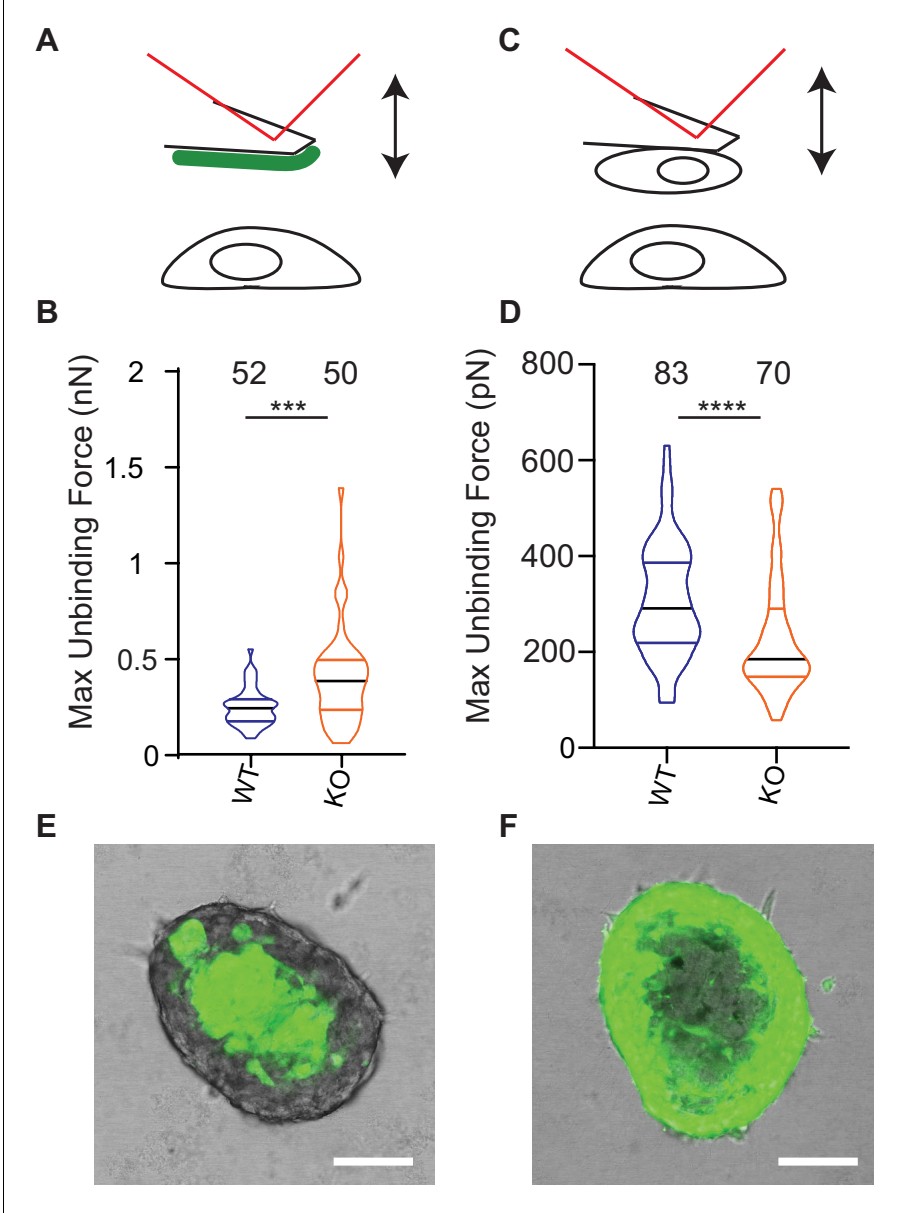

**Figure 8.** Deletion of Elkin1 modulates cell binding forces. (**A**) Diagram of AFM analysis of cell-matrix unbinding forces. (**B**) These data demonstrated a significant increase in the maximum unbinding force to separate cells from LM511, after a 2 s contact time (Mann-Whitney U test, WT = 52, KO = 50 force-distance curves, ***p=0.001) (**C**) Diagram of AFM analysis of cell-cell unbinding forces (**D**) A significantly higher force was required for the unbinding of WM266-4 WT cells in comparison with the Elkin1-KO cells after a 2 s contact time (Mann-Whitney U test, WT = 70, KO = 83 force-distance curves, ****p<0.0001). (**E–F**) Representative confocal images taken from an orthogonal slice through chimeric spheroids formed over 72 hr from equal numbers of (**E**) WT-GFP cells and unlabelled Elkin1-KO cells or (**F**) unlabelled WT cells and Elkin1-KO-GFP cells. In both cases the spheroid organises with the WT cells in the core and the Elkin1-KO cells in the periphery. Scale bars = 100 μm. Similar observations made for 10 WT-GFP:Elkin-KO and 13 WT:Elkin-KO-GFP chimeric spheroids. See *Figure 8—source data 1* for further details on cell binding forces. See *Figure 8—figure supplement 1* for spheroid chimera rescue experiments using Elkin1 variants.

The online version of this article includes the following source data and figure supplement(s) for figure 8:

**Source data 1.** Source data for AFM measurements of cell binding forces.
**Figure supplement 1.** Overexpression of Elkin1-L210F does not rescue the partitioning phenotype in chimeric spheroids.

*Poole et al., 2014*; *Rocio Servin-Vences et al., 2017*) (though PIEZO2 is less sensitive to membrane stretch than PIEZO1 [*Wang et al., 2019*]) and TRPV4 responds only to substrate deflections in mammalian cells (*Rocio Servin-Vences et al., 2017*). These data suggest that the expression of distinct MA channels allows cells to distinguish between stimuli arising from cell-generated forces at cell-substrate contacts *versus* exogenous stimuli that stretch the membrane or compress the cell. Such differential activation profiles may reflect multiple roles for distinct MA channels during tumour development and metastasis.

During metastasis, cells first dissociate from the primary tumour and then, as they migrate through the body, the cells encounter environments of differing dimensionality and degree of confinement (*Paul et al., 2017*; *van Helvert et al., 2018*). Disruption of *Elkin1* expression altered the confined and unconfined migration of cells in *quasi*-1D and 2D environments. Our results stand in contrast with the effect of PIEZO1 on melanoma cell migration: PIEZO1 knockdown reduced the speed of cell migration in confined environments, but had no effect on unconfined cell migration (*Hung et al., 2016*). In other tumours, PIEZO2 knockdown in a breast cancer cell line (BrM2), inhibited the cells' ability to enter confined spaces, but did not influence the speed of migration of the confined cells (*Pardo-Pastor et al., 2018*). These data suggest that distinct MA channels mediate separate mechanical transduction pathways regulating different aspects of tumour cell migration.

In addition to the Elkin1-dependent changes in migration, we report that Elkin1-KO cells exhibited increased dissociation from organotypic spheroids embedded in collagen gels. The process of cell dissociation from a spheroid mass is defined by a complex interplay between the strength of cell-cell adhesions, the contractility of the cells at the edge of the spheroid and the physical characteristics of the surrounding matrix (*Ahmadzadeh et al., 2017*). Our data demonstrate that Elkin1 expression regulates cell-substrate and cell-cell binding forces. In addition, the partitioning of cells within the organotypic spheroids was dependent on Elkin1. We therefore propose that Elkin1-dependent expression regulates the balance between cell-cell and cell-matrix adhesion; ablation of the protein increases invasion of surrounding matrix (by reducing cell-cell adhesions and facilitating cellular dissociation) and slows migration (by increasing cell-substrate binding). Previous studies of the role of TRPV4 in breast cancer development have demonstrated that overexpression of TRPV4 can lead to a decrease in E-cadherin expression, driving EMT in this cancer type (*Lee et al., 2017*). This TRPV4-dependent switch in E-cadherin expression may also lead to increased dissociation from the tumour mass. However, in contrast to Elkin1, increased levels of TRPV4 lead to an increase, rather than a decrease in cell migration (*Lee et al., 2016b*; *Lee et al., 2017*). A similar phenomenon (decreased cell-cell binding forces, facilitated dissociation from organotypic spheroids and decreased cell migration) has been reported in melanoma cells overexpressing the Na$^+$/H$^+$ exchanger, NheI (*Hofschröer et al., 2017*). This exchanger locally modulates the pH of the cellular environment and is not known to be mechanically responsive.

Given that Elkin1 is required for MA currents that are separable from the PIEZO channels and that Elkin1-dependent currents are activated at the cell-substrate interface, these channels may transduce distinct mechanical inputs. The PIEZOs may be acting to sense confinement (*Hung et al., 2016*; *Pardo-Pastor et al., 2018*) and the overall mechanical status of the cells (*Rocio Servin-Vences et al., 2017*), while Elkin1-dependent mechanoelectrical transduction may encode information about the mechanical nature of the cells' microenvironment thus facilitating modulation of cell-cell *versus* cell-substrate binding. Such integration of multiple mechanoelectrical transduction pathways could engender cells with a tuneable and diverse repertoire of mechanical sensing.

## Materials and methods

Please refer to *Supplementary file 2*: Key Resources table for details of resources used and created for this study.

### Cell culture

All melanoma cell lines (WM266-4, WM115, and A375) were obtained from the ATCC and cultured in complete Minimum Essential Medium Eagle (MEME) or High Glucose DMEM supplemented with 10% FBS and 1% Penicillin/Streptomycin. MEME was additionally supplemented with 1% l-glutamine. HEK-293T and HEK-293T P1KO (*Lukacs et al., 2015*) (a gift from A. Patapoutian) were cultured in DMEM medium, 10% FBS and 1% Penicillin/Streptomycin. All cultures were maintained at

37°C, 5% $CO_2$. Fugene HD (Promega) or polyethyleneimine (PEI, Sigma, MW 600–800 kDa) was used to transfect cells at the following ratios (w/w): DNA:Fugene at 1:3 and DNA:PEI at 1:4. WM266-4 cells were transduced with pRRLSIN.cPPT.PGK-GFP.WPRE lentivirus (a gift from Ian Alexander, originally from Didier Trono/Inder Verma, Addgene plasmid #12252), in which a human PGK promoter drives EGFP expression. The virus was produced in HEK-293T cells following PEI transfection of the lentiviral backbone together with packaging plasmid (psPAX2) and the vesicular stomatitis virus (VSV-G) envelope. To create cell lines with stably-integrated DNA for spheroid rescue experiments, plasmids encoding Elkin1 and GFP were linearised using ScaI. Linearised DNA was transfected into WM266-4 3C6 (Elkin1 KO clone) using Fugene (as above). Genetically modified cells (virally transduced or stably expressing Elkin1) were isolated in FACS buffer (2 mM EDTA, 2% FBS, 2% Penicillin/ Streptomycin in PBS) using the BD FACS Jazz (low pressure sorting with a 100 µm nozzle at 17 psi, 4°C). For all experiments to be conducted within 6 hr of cell collection, cells were released from culture flasks using enzyme-free cell dissociation buffer (Sigma-Aldrich). Cell lines were authenticated using STR profiling and checked for mycoplasma contamination by CellBank Australia.

## Proteomics

To prepare samples for mass spectrometry, peptides were prepared from cultured WM266-4 cells using the Pierce Mass Spec Sample Prep Kit for Cultured Cells as per manufacturer's instructions (ThermoFisher). LysC and Trypsin were used to generate peptides from 1 mg of total protein. Peptides were separated using isoelectric focussing (IEF) in immobilized pH gradient (IPG) gel strips into six separate fractions, as previously described (*Eravci et al., 2014*). Desalted peptides of these fractions were then separated on an 8–60% acetonitrile gradient (240 min) with 0.1% formic acid at a flow rate of 200 nL/min using the EASY-nLC II system (Thermo Fisher Scientific) on in-house manufactured silica microcolumns packed with the ReproSil-Pur C18-AQ 3 µm resin. A Q Exactive plus mass spectrometer (Thermo Fisher Scientific) was operated in the data dependent mode with a full scan in the Orbitrap followed by top 10 MS/MS scans using higher-energy collision dissociation (HCD). Analysis of MS and MS/MS spectra was performed using MaxQuant software (version 1.5.1.2) and proteins were identified by searching against the human reference proteome database UP000005640 from Uniprot.

## Molecular biology

The sequences of all primers used for this study are listed in *Supplementary file 2*. mRNA was isolated from WM266-4 cells using the RNEasy kit as per manufacturer's instructions (Qiagen). First strand cDNA synthesis was carried out using 200 ng of isolated mRNA, random primer mix (New England Biolabs) and M-MuLV reverse transcriptase (New England Biolabs). The resulting samples were used for qPCR analysis (see below) and as a template to amplify *hs*Elkin1-iso1/3 cDNA. For electrophysiology experiments, these sequences were cloned into pRK5 with a cistronic eGFP driven by an internal IRES sequence. For the purposes of live cell imaging, *hs*Elkin1-iso1/iso3 sequences were fused to mGFP using the same vector. miRNA knockdown reagents were generated using the BLOCK-iT Pol II miR RNAi Expression Vector, as per manufacturer's instructions (Invitrogen). Briefly, complementary ssDNA sequences (Eurofins, Belgium) that encode the miRNA of interest were annealed and cloned into the pcDNA 6.2GW-EmGFP vector. Three distinct miRNA sequences were generated for each target gene and assembled in series within a single plasmid to generate the final construct for knock-down experiments. The Golgi network was detected using CellLight Golgi-RFP, BacMam 2.0 (ThermoFisher). Sequences were analysed using the T-Coffee multiple sequence alignment server (coffee.crg.cat) and formatted using Boxshade (https://embnet.vital-it.ch/software/ BOX_form.html).

## CRISPR/Cas9 gene editing

To create plasmid constructs for gene editing, guide RNAs (gRNAs) were designed using the online CRISPR design tool (Zhang Lab, MIT - http://crispr.mit.edu/). Single stranded DNA oligos were obtained from Eurofins (Belgium), annealed and cloned into the pSpCas9n(BB)−2A-GFP plasmid (a gift from Feng Zhang, Addgene plasmid # 48140) (*Ran et al., 2013*). Four plasmids were generated, each with gRNAs that targeted sequences in intron 7 or exon 9 of the *Elkin1* gene. These constructs were transfected into either WM266-4 or A375 cells. Isolation of clonal populations was a two-step

process. First, a population of transfected cells was initially isolated based on GFP expression: 24–48 hr post-transfection cells were resuspended in FACS buffer (2 mM EDTA, 2% FBS, 2% Penicillin/Streptomycin in PBS) and cells expressing GFP were collected using the BD FACS Jazz low pressure sorting with a 100 µm nozzle at 17 psi, 4°C. Following expansion of this population, cells were sorted a second time (as above) as single cells in individual wells of a 96 well plate. Single-cell clones were initially cultured in a media comprising of 50% complete media/50% conditioned media for 7–14 days until cells had started to divide and form colonies.

### Genomic PCR

Edited clones were screened for genomic deletions covering the *Elkin1* gene using PCR. Primers for screening CRISPR edited clones were designed using Primer-BLAST (NCBI) to span the ~2.5 kb deleted region of the gene. Genomic DNA (gDNA) was extracted from cells using the Illustra Genomic Prep Mini Spin kit (GE Life Sciences), as per manufacturer's instructions. Genomic PCR was conducted using 10 ng of template DNA and Hot Start *Taq* DNA polymerase (New England Biolabs).

### Quantitative PCR

Primers and probes to analyse *Elkin1* and *PIEZO1* transcript levels were designed using PrimerQuest Design Tool (IDT, Singapore) and a predesigned assay was used to detect levels of HPRT1 (housekeeping gene) (Integrated DNA Technologies, Singapore). The reaction underwent 40 cycles. Using the difference in cycle threshold ($\Delta Ct$), the fold change in expression ($2^{-\Delta\Delta Ct}$) was calculated and compared to a control sample.

### Micropillar array fabrication

Positive masters and pillar array casting were described previously (*Poole et al., 2014*). Briefly, positive silicon masters were silanised using vapour phase Trichloro(1H,1H,2H,2H-perfluorooctyl) silane (Sigma-Alrich) for 16 hr. Negative masters were cast from this substrate in polydimethylsiloxane (PDMS) (Sylgard 184, Dow Corning), mixed at a ratio of 1:10 and cured at 110°C for 15 min. Negative masters were silanised as above and used to cast pillar arrays. Arrays were coated with degassed PDMS (1:10) and left for 30 min. A thickness two coverslip activated with oxygen plasma generated using a low pressure Zepto plasma system (Diener, Germany) was placed over the still liquid PDMS. Pillar arrays were cured for 1 hr at 110°C. Pillar arrays were activated using the oxygen plasma system and either cells were directly seeded onto this activated surface or arrays were first functionalised by coating with 10 µg/ml LM-511 (BioLamina, Sweden) for one hour at 37°C. Cells were seeded at a concentration of $2 \times 10^4$ cells/mL in complete media and incubated overnight.

### Electrophysiology

Whole-cell patch pipettes were prepared from thick-walled filamented glass (Harvard Apparatus, USA) using a pipette puller fitted with a box filament (P-1000, Sutter Instruments, USA). Pipettes were heat-polished with a homemade micro-forge to give a final resistance of 3 MΩ – 6 MΩ. Pipettes were filled with a solution containing 110 mM KCl, 10 mM NaCl, 1 mM MgCl$_2$, 1 mM EGTA and 10 mM HEPES (pH 7.3). Extracellular solutions contained 140 mM NaCl, 4 mM KCl, 2 mM CaCl$_2$, 1 mM MgCl$_2$, 4 mM glucose and 10 mM HEPES (pH 7.4). Whole-cell patch-clamp data was obtained on either a Zeiss 200 inverted microscope and an EPC-10 amplifier in combination with Patch-master software or a Nikon T*i*-E inverted microscope and an Axopatch 200B with pClamp 10 software. Data were analysed using either Fitmaster software (HEKA Electronik GmbH, Germany) or Clampfit software (Molecular Devices, USA). Pipette and membrane capacitance were compensated and to minimise voltage errors, series resistance was compensated by at least 60%. Mechanically-activated currents were recorded at a holding potential of −60 mV.

Mechanical stimuli were applied by serially deflecting an individual pilus using a blunt, heat-polished pipette (tip diameter approx. 2 µm) driven by a MM3A-LS nanomanipulator (Kleindiek Nanotechnik, Germany). Multiple stimuli ranging between 1 nm–1 µm were applied with a delay between each stimulus of at least 10 s. To quantify the stimulus, bright-field images of the deflected pilus were taken before and during stimulation using a 40x/0.6 NA objective. The centre of each pilus was calculated off line by applying a 2D-gaussian fit of intensity values (Igor, Wavemetrics, USA) and the

magnitude of pillar deflection calculated from successive images by comparing the difference in the calculated centre point.

## Transswell assays

Transwell assays were conducted using the HTS FluoroBlok Multiwell Insert System (Corning) with an 8.0 µm pore size. The bottom surface of the membrane was coated with laminin (10 µg/ml) for 3 hr at 37˚C. Membranes were then blocked for 1 hr using 2% polyvinylpyrolidone. The bottom well was filled with 500 µl media and $1 \times 10^4$ cells were added to the top well. Cells were allowed to transmigrate for 16 hr before the bottom side of the membrane was fixed with 4% PFA. In order to quantify the number of cells that had transmigrated, the fixed sample was permeabilised using 0.1% TX-100 for 5 min at room temperature and nuclei were stained using Hoechst at a concentration of 0.1 µg/ml. Cells were imaged using epifluorescence imaging on an inverted microscope (Zeiss Axiovert 200 or Leica DMIL) fitted with a 10x/0.22 NA objective and standard filters; 4 regions of each filter were imaged. The 'Analyze Particles' function of ImageJ was used to count cells within each field. Data from each independent experiment were normalized to the average number cells that were counted on the underside of each filter for triplicate controls.

## 2D migration assay

µ-Slide 8-well dishes (ibidi, Germany) were coated with either 10 µg/ml LM-511 (BioLamina, Sweden), or 0.1% (w/v) poly-L-lysine (PLL) (Sigma-Aldrich, USA). Approximately 2500 cells were added to each well, and after 2 hr, nuclei were labelled by adding Hoechst dye at a concentration of 0.1 µg/ml. Cell migration was monitored for 16 hr using live-cell microscopy (see below). Data were obtained from three experiments. Data were confirmed by repeating this assay with cells expressing cytoplasmic GFP, using the same imaging conditions.

## quasi-*1D migration assay*

Microcontact-printing (*von Philipsborn et al., 2006*) was used to pattern substrates for *quasi*-1D experiments. Microcontact-printing stamps (*Chiang et al., 2011*) were incubated with 10 µg/ml LM511 and 1 µg/ml of goat anti-mouse IgG antibody labelled with AlexaFluor 594 (to visualize patterns) for 30 min at 37 ˚C. Stamps were rinsed with ultrapure water, dried with inert gas, then stamped on clean coverslips activated using oxygen plasma. Coverslips were incubated with 1 mg/mL PLL-g-PEG (SuSoS, Switzerland) in PBS, at room temperature for 30 min, to block unprinted regions. Cells ($3 \times 10^4$ cells/mL) were seeded on each substrate and nuclei were labelled by adding 0.1 µg/ml Hoechst. Cells were monitored using live-cell microscopy for 12 hr and data were obtained from at least 50 cells across three experiments. We only included in our data the periods of movement that correspond to when the cell was attached and elongated on the patterned region. To determine whether cells were confined/unconfined we analysed the cell shape. Cells with an aspect ratio of 2:1 or more were classed as confined and the number of confined vs non-confined cells was determined by sampling still images at five points during each experiment.

## Microscopy

### Quasi-*1D and 2D migration*

The Nikon Eclipse T*i*-E inverted microscope fitted with a 20x/0.5 NA objective and the Perfect Focus System was used for live imaging of migrating cells at 37˚C, 5% $CO_2$. Phase-contrast (exposure 200 ms) and epifluorescent images (exposure 850 ms) were obtained every 5 min. Data was collected using the NIS elements software. Data of 2D and *quasi*-1D migration were analysed using the ImageJ plugin TrackMate (version 1.51n) (*Tinevez et al., 2017*) with the parameters: nuclear diameter ~13 µm, simple Linear Assignment Problem (LAP) tracker, linking distance and gap-closing maximum distance of 20 µm. Migration tracks were discarded if the track was shorter than 5 h. Mean speed was calculated from the entire length of the track. The Euclidean distance was calculated as $\sqrt{(x)^2 + (y)^2}$, where $x$ (= $x_B - x_A$) and $y$ (= $y_B - y_A$) are the differences in the $x$ and $y$ axes, respectively, and $x_A$, $y_A$ is the coordinate of the track origin and $x_{B,}, y_B$ is the coordinate of the final point (taken at 5 h for each track). Data were excluded if cells became necrotic during the imaging.

### Imaging Elkin1 variants

For live imaging, cells were plated onto 35 mm glass bottom dishes (ibidi GmbH, Germany) and transiently transfected using Fugene HD (Promega, USA) 24 hr prior to imaging. All cultures were maintained at 37°C and 5% $CO_2$ for the duration of the experiment. Confocal microscopy was performed on a Zeiss 880 equipped with a Plan-Apochromat 63×/1.4 NA Oil DIC M27 objective and an Airyscan detector. TIRF imaging was performed using the Zeiss Elyra system fitted with a Plan-Apochromat 100×/1.46 NA Oil objective and an Andor iXon 897 EMCCD camera. In both cases, samples were illuminated using 488 nm, 561 nm or 633 nm lasers, and all data was collected using the Zen software (Zeiss).

## 3D dissociated cell migration assay

To assay individual cells in collagen gels, a cell suspension ($1.25 \times 10^6$ cells/mL in complete media) was mixed with 1.5 mg/mL rat tail Collagen I (total volume 100 μL) and gelated at 37°C, 5% $CO_2$ for 15 min. Isolated cells in collagen gels were imaged using a Leica SP8 DLS fitted with a 10x/0.4 NA objective every 10 min for 14 hr. All cultures were maintained at 37°C and 5% $CO_2$.

## Organotypic spheroid assay

Both WT and Elkin1-KO clones were virally transduced with a construct encoding eGFP to enable visualisation of cells. For spheroid formation, $1 \times 10^3$ cells were seeded onto Ultra-low attachment 96-well plates (Corning, USA) in complete MEME media and formed spheroids following 72 hr of incubation. Composite spheroids were created by mixing $5 \times 10^2$ of each cell-type to give $1 \times 10^3$ cells total. Collagen gel mixtures were made from 1.5 mg/mL – 2 mg/mL rat tail Collagen I (Corning, USA) in 10 mM NaOH, 1 × PBS and complete media on ice. A base collagen gel was formed in each well of a glass-bottom 96-well plate (Greiner Bio-one, Austria) by incubating 30 μL of the collagen mixture at 37°C, 5% $CO_2$ for 6 min. The collagen gel mixture containing the spheroid was then applied to the base gel and underwent gelation at 37°C, 5% $CO_2$ for 15 min. Cells in collagen gels were cultured under 200 μL of complete media. Spheroids and invasive cells were imaged at 24, 48, 72 hr using a Leica SP8 DLS fitted with a 10x/0.4 NA objective.

## Analysis of cells in collagen gels

Imaris 9.1.2 software (Bitplane AG, Zurich, Switzerland) was used to segment cells by creating surfaces with a filter of 200 μm³ to discard cell debris. For organotypic spheroid assay analysis, cell surfaces were filtered by centre of image mass *versus* z depth, where the z value was determined by orientating the 3D image stacks to the plane of view showing the cells on the glass and cells attached to the glass were excluded from analysis. Cells from the 3D dissociated cell migration assay were tracked using autoregressive motion, applying a threshold of 1500s to filter track duration. Intensity, morphological and tracking data were then exported and further analysed using GraphPad Prism software (La Jolla, CA, USA).

## Immunoblotting and cell-surface biotinylation

To detect Elkin1 variants at the plasma membrane, the cell surface fraction was biotinylated and subsequently isolated following established protocols (Tarradas et al., 2013). Briefly, HEK-293T P1KO cells attached to PLL-coated culture dishes were transiently transfected with plasmids encoding GFP-tagged Elkin1 variants. After 24 hr the cell surface fraction was labelled, on ice, with freshly prepared 2.5 mg/ml EZ-link Sulfo-NHS-LC-LC-biotin (21338, ThermoFisher Scientific) in DPBS containing $Ca^{++}$. After quenching with 100 mM glycine, cells were lysed as above using RIPA buffer containing protease inhibitor cocktail. A portion of this lysed sample was reserved as the 'input' sample. The biotinylated fraction was then isolated using NeutrAvidin Ultralink Resin (53150, ThermoFisher Scientific). After recovery from the NeutrAvidin beads, samples were prepared as for gel electrophoresis by mixing with Bolt LDS sample buffer and Bolt reducing agent (B0007 and B0009 respectively, ThermoFisher Scientific). Samples were separated on a 10% Bolt Bis-Tris Plus gel (NW00102BOX, ThermoFisher Scientific), transferred to a PVDF membrane and subjected to standard antibody detection. GFP-fusion proteins were detected using rabbit polyclonal anti-GFP (SAB4301138, Sigma-Aldrich, 1:1000) and HRP-linked anti-rabbit IgG (7074, Cell Signaling Technologies, 1:1000).

## AFM analysis of cell binding forces

AFM analysis of cell-binding forces was conducted using a JPK Nanowizard, fitted with a CellHesion module (JPK Instruments AG). To measure cell-LM511 interaction forces, cantilevers (MLCT-O10, Bruker) were first activated using oxygen plasma generated using a low-pressure Zepto plasma system (Diener, Germany) and then incubated in a droplet of LM511 (20 µg/ml) for 1 hr. To measure cell-cell interaction forces, cantilevers were coated with wheat germ agglutinin (L4895, Sigma Aldrich). Before collecting data, the sensitivity and spring constant of each cantilever was determined using in-built routines in the JPK software. To measure cell-LM511 binding forces, the following settings were used: speed 2 µm/sec, set point 500 pN, with contact times of 2 s held at constant height. At least five force-distance curves were collected from at least 10 cells per condition (multiple force-distance curves were not collected successively on each cell to minimise adaptive changes). To measure cell-cell binding forces the CellHesion module was used to move the stage through an extended 100 µm pulling range. To attach a cell to the cantilever, cells in suspension were added to a sample dish containing adherent cells, the calibrated cantilever was positioned over an unattached cell and carefully brought into contact. After 10 s attachment time the cantilever was slowly withdrawn from the surface, with a single cell attached. To collect data the cantilever was positioned over an adherent cell and data collected with the following parameters: speed 5 µm/sec, set point 250 pN, contact time 2 s held at a constant height. At least five force-distance curves were collected from at least 10 different cell-cell interaction pairs. Force-distance curves were analysed to determine the maximum unbinding force during cantilever retraction.

## Statistical analysis

All data were analysed using Prism seven or Prism 8 (GraphPad). Normality was determined with D'Agostino-Pearson omnibus normality test. Normally distributed data were analysed using parametric statistical tests. All *t*-tests were two-tailed and when there were significant differences in the variance, Welch's correction was used. Data from qPCR experiments and migration assays were analysed using a Kruskal-Wallis one-way ANOVA test with Dunn's multiple comparisons. Stimulus-response plots were generated by binning data by stimulus size and averaging within bins for each cell, then across cells. Ordinary two-way ANOVA with Sidak's multiple comparisons was used to analyse stimulus-response plots. The transwell migration assay data were normalised against the parental clone and analysed using a parametric one-way ANOVA test with Tukey's multiple comparisons. Data from organotypic spheroid experiments were averaged for each spheroid and then compared with one-way ANOVA with Tukey's multiple comparisons.

## Acknowledgements

The authors would like to thank Liana Kosizki, Heike Thrainhart and Feyza Colakoglu for technical assistance and Matt Baker and Efrosini Deligianni for advice on Greek terminology. This work was supported by a Cecile Vogt Fellowship from the MDC to KP, a Collaborative Research Center Grant from the Deutsche Forschungsgemeinschaft, SFB958 (Project A09, to KP and GRL; project Z03 to ME, CW), Humboldt Research Fellowship to MM, an Australian Government RTP scholarship to AP an NHMRC project grant to KP, BM and MB (APP1138595) and an NHMRC Principal Research Fellowship to BM (APP1135974). Live cell migration, TIRF and confocal imaging was conducted in the Biomedical Imaging Facility and flow cytometry at the Biological Resources Imaging Laboratory both at the Mark Wainwright Analytical Centre, UNSW Sydney. Thanks to Joshua Chou (University of Technology, Sydney) and Michael Higgins (University of Wollongong) for the use of their AFM instruments. The Genotype-Tissue Expression (GTEx) Project was supported by the Common Fund of the Office of the Director of the National Institutes of Health, and by NCI, NHGRI, NHLBI, NIDA, NIMH, and NINDS. The GTEx data described in this manuscript were obtained from the GTEx Portal on 20/08/18.

## Additional information

### Funding

| Funder | Grant reference number | Author |
| --- | --- | --- |
| National Health and Medical Research Council | APP1138595 | Boris Martinac<br>Maté Biro<br>Kate Poole |
| Deutsche Forschungsge-meinschaft | SFB958, A09 | Gary R Lewin<br>Kate Poole |
| National Health and Medical Research Council | APP1135974 | Boris Martinac |
| Deutsche Forschungsge-meinschaft | SFB958, Z03 | Murat Eravci<br>Christoph Weise |
| Humboldt Foundation | Postdoctoral Fellowship | Mirko Moroni |
| Max Delbruck Center | Cecile Vogt Fellowship | Kate Poole |
| Department of Education, Australian Government | RTP scholarship | Amrutha Patkunarajah |

The funders had no role in study design, data collection and interpretation, or the decision to submit the work for publication.

### Author contributions

Amrutha Patkunarajah, Formal analysis, Investigation, Methodology; Jeffrey H Stear, Supervision, Investigation, Methodology; Mirko Moroni, Supervision, Investigation; Lioba Schroeter, Jedrzej Blaszkiewicz, Charles D Cox, Carina Fürst, Oscar Sánchez-Carranza, María del Ángel Ocaña Fernández, Raluca Fleischer, Investigation; Jacqueline LE Tearle, Investigation, Methodology; Murat Eravci, Funding acquisition, Investigation; Christoph Weise, Funding acquisition, Methodology; Boris Martinac, Supervision, Funding acquisition; Maté Biro, Supervision, Funding acquisition, Methodology; Gary R Lewin, Conceptualization, Resources, Supervision, Funding acquisition, Project administration; Kate Poole, Conceptualization, Resources, Formal analysis, Supervision, Funding acquisition, Investigation, Methodology, Project administration

### Author ORCIDs

Boris Martinac (iD) http://orcid.org/0000-0001-8422-7082
Maté Biro (iD) http://orcid.org/0000-0001-5852-3726
Gary R Lewin (iD) https://orcid.org/0000-0002-2890-6352
Kate Poole (iD) https://orcid.org/0000-0003-0879-6093

### Decision letter and Author response

Decision letter https://doi.org/10.7554/eLife.53308.sa1
Author response https://doi.org/10.7554/eLife.53308.sa2

## Additional files

### Supplementary files

- Supplementary file 1. Proteins identified in WM266-4 cells.
- Supplementary file 2. Key resources table.
- Transparent reporting form

### Data availability

All data generated or analysed during this study are included in the manuscript and supporting files. Source data file has been provided for figures 1, 3, 4, 6, 7, 8. Proteomics data provided as Supplementary file 1.

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
