## [Decision Letter]

**Acceptance summary:**

PIEZO channels are the primary mechanically activated ion channels in higher eukaryotes including humans. This study utilizes electrophysiology and cell biology assays to provide compelling evidence that TMEM87a (renamed Elkin1) is a major component of a PIEZO1 independent mechanotransduction pathway.

**Decision letter after peer review:**

Thank you for submitting your article "TMEM87a/Elkin1-dependent mechanoelectrical transduction modulates melanoma cell adhesion and migration" for consideration by *eLife*. Your article has been reviewed by Jonathan Cooper as the Senior Editor, a Reviewing Editor, and three reviewers. The following individuals involved in review of your submission have agreed to reveal their identity: Jerome Jacques Lacroix (Reviewer #3).

The reviewers have discussed the reviews with one another and the Reviewing Editor has drafted this decision to help you prepare a revised submission.

Summary:

In this manuscript, Stear et al. describe the transmembrane protein Elkin1 as necessary and sufficient to confer mechanically activated currents to otherwise mechanically-insensitive cells. Further, they relate cell adhesion and migration properties to the expression of Elkin1. As such Elkin1 is a strong candidate protein for a novel mechanically-activated ion channel and a regulator of cell adhesion/migration. Overall, the manuscript is very well written and the experiments appear to be performed and analyzed at a high standard. The identification of Elkin1 as a necessary and sufficient protein for mechanotransduction pathway is a significant and novel finding.

Essential revisions:

1) The authors claim that their findings directly show that Elkin1-dependent mechanically activated current modulates cellular adhesions and regulates cell migration and cellular interactions, however this is vastly overstated and is not supported by any data. The experiments presented here demonstrate that the expression of Elkin1, not the Elkin1-dependent mechanical currents, modulates the cellular phenotypes as none of the experiments are designed to test the effect of current on cellular properties. Below I am listing all specific claims that need correction:

- The current title of the paper "…mechanoelectrical transduction modulates…" makes this causative connection and needs to be changed.

- Abstract: "We therefore conclude that Elkin1 supports a Piezo1-independent mechanoelectrical transduction pathway that modulates cellular adhesions…". The authors should replace the word "that", which incorrectly implicates causation, with "and".

- Discussion section: "Elkin-1 dependent MA currents altered the confined and confined migration…". The experimental design is to alter the expression pattern of Elkin1, not the MA currents, and test for changes in cellular migration. The authors should rephrase.

- Discussion section: "Our data demonstrate that Elkin1 activity regulates cell-substrate…." Again, the experiment does not allow this conclusion. The word "activity" should be replaced with "expression".

- Discussion section: "We therefore propose that Elkin1-dependent mechanotransduction regulates the balance…" The word "mechanotransduction" should be replaced with "expression".

- Discussion section: "To our knowledge, this study represents the first direct demonstration of a role for mechanoelectrical transduction in regulating the physical interactions…". This statement should be removed entirely.

- Discussion section: The first two sentences of this paragraph should be removed.

2) The differences between the mouse and human Elkin1 sequences and the related differences in their electrophysiological activity was used to identify two residues key for the ion channel's function. This is a very interesting finding, but it leaves a question unanswered about the role of the mouse Elkin1 given the lack of robust ionic currents. Can the authors comment on this?

3) Why do WT and Elkin1-knockout cells separate from each other in the spheroid assay? Is the ion channel activity of the protein important for this effect, or is it related to other aspects of its expression? It would be important to do this assay with cells expressing the L210F or the G231N mutant channels to clarify this issue.

4) The authors seem to suggest that Elkin1 knockout elicits an epithelial-to-mesenchymal transition. However, as the authors point out, the WT melanoma cells which express Elkin1 already show some mesenchymal characteristics. Moreover, the KO cells show a propensity for dissociating from the spheroid (consistent with an EMT phenotype), but also slower migration (contradictory to cancer cells that have undergone EMT). Given these ambiguities, it is better to eliminate the data on EMT markers (Figure 8A and subsection “Deletion of Elkin1 changes expression of epithelial and mesenchymal markers and modulates cell binding forces”) from the present study.

5) Hoechst labeling of live cell experiments may compromise cell health, especially over long time-lapse imaging. The authors should include details on live-cell imaging (e.g. interval during images and duration of illumination for each time point) and ideally also confirm the results with cells labeled with a different fluorophore that is gentler on cells, e.g. GFP labeled cells.

6) The authors conclude that knockdown of Elkin1 in WM266-4 cells results in reduction in MA currents, however in Figure 2A the final data point on the graph is nearly identical between WT and cells treated with Elkin1 miRNA. This supports the idea that Elkin1 may be a modulator of a MA channel and knockdown in Elkin1 expression results in a rightward shift in the sensitivity of the MA currents. The authors should address this possibility and state more clearly within the discussion that the role of Elkin1 in this mechanotransduction pathway is still not known.

---

## [Author Response]

Essential revisions:1) The authors claim that their findings directly show that Elkin1-dependent mechanically activated current modulates cellular adhesions and regulates cell migration and cellular interactions, however this is vastly overstated and is not supported by any data. The experiments presented here demonstrate that the expression of Elkin1, not the Elkin1-dependent mechanical currents, modulates the cellular phenotypes as none of the experiments are designed to test the effect of current on cellular properties. Below I am listing all specific claims that need correction:

In light of these reviewer comments, particularly the fact that we have tested the impact of changes in Elkin1 expression rather than activity, we have made all of the appropriate changes suggested by the reviewers. Please find specific comments to the individual points raised in response to the reviewers’ suggestions below.

- The current title of the paper "…mechanoelectrical transduction modulates…" makes this causative connection and needs to be changed.

The title has been changed to remove the causative connection and now appears as:

TMEM87a/Elkin1, a component of a novel mechanoelectrical transduction pathway, modulates melanoma adhesion and migration

- Abstract: "We therefore conclude that Elkin1 supports a Piezo1-independent mechanoelectrical transduction pathway that modulates cellular adhesions…". The authors should replace the word "that", which incorrectly implicates causation, with "and".

This modification has been made.

- Discussion section: "Elkin-1 dependent MA currents altered the confined and confined migration…". The experimental design is to alter the expression pattern of Elkin1, not the MA currents, and test for changes in cellular migration. The authors should rephrase.

We have rephrased this sentence as “Disruption of Elkin1 expression altered the confined and unconfined migration…”.

- Discussion section: "Our data demonstrate that Elkin1 activity regulates cell-substrate…." Again, the experiment does not allow this conclusion. The word "activity" should be replaced with "expression".

This modification has been made.

- Discussion section: "We therefore propose that Elkin1-dependent mechanotransduction regulates the balance…" The word "mechanotransduction" should be replaced with "expression".

This modification has been made.

- Discussion section: "To our knowledge, this study represents the first direct demonstration of a role for mechanoelectrical transduction in regulating the physical interactions…". This statement should be removed entirely.

We have removed this sentence from the manuscript.

- Discussion section: The first two sentences of this paragraph should be removed.

We have removed these two sentences from the manuscript.

2) The differences between the mouse and human Elkin1 sequences and the related differences in their electrophysiological activity was used to identify two residues key for the ion channel's function. This is a very interesting finding, but it leaves a question unanswered about the role of the mouse Elkin1 given the lack of robust ionic currents. Can the authors comment on this?

We report here a role for the human Elkin1 in melanoma migration and cellular dissociation. It is important to note that melanoma progression is quite distinct in mice versus humans. In particular, the microenvironment in which primary tumours develop is distinct, with human melanoma arising in the epidermis and mouse largely in the dermis (Sun et al., 2019, Nat. Commun.). As such, the process of cell dissociation and migration from the primary tumour will involve interactions with different microenvironments.

We agree with the reviewers that the differences between the mouse and human Elkin1 electrophysiological activity are very interesting. We would like to highlight the fact that expression of the mouse variant (and the human with the mouse residues introduced into the sequence) is still associated with the de novo appearance of mechanically activated currents in cells, though the currents have smaller amplitudes. As such, differences in expression levels or the microenvironment may positively impact *mm*Elkin1-dependent channel activity in situ.

Given the lack of information about this protein in either organism we have refrained from adding additional speculation about a broader set of functions for Elkin1 outside the cells that we have studied here. We feel that this is justified by the fact that in distinct cells and tissues other MA channels have been shown to exhibit quite different current amplitudes and thresholds depending on a number of factors, including the microenvironment (Bavi et al., 2019, ACS Nano.; Chiang et al., 2011) and the co-expression of modulatory protein partners (Poole et al., 2014; Qi et al., 2015, Nat Commun.; Zhang et al,. 2017).

The characterisation of the impact of Elkin1 in other cells and tissues in the mouse is beyond the scope of this paper but we are currently generating mutant mice in order to specifically study this question.

3) Why do WT and Elkin1-knockout cells separate from each other in the spheroid assay? Is the ion channel activity of the protein important for this effect, or is it related to other aspects of its expression? It would be important to do this assay with cells expressing the L210F or the G231N mutant channels to clarify this issue.

We thank the reviewers for this suggestion and have conducted revision experiments to address this point. Given the time that it takes for the cells to form spheroids (days) this experiment required stably transfected cells. As detailed in the revised Materials and methods, we linearised the relevant plasmids (co-expressing GFP with either Elkin1-iso3 or Elkin1-iso3-L210F) in order to track stable integrants. Linearised plasmid was introduced into one of the Elkin1-KO clones. After 1 week in culture we sorted the population of green cells and re-established the cell population. Chimeric spheroids were then formed with the parental Elkin1 KO clone. We found that those cells expressing the WT version of Elkin1 were more likely to be found in the core of the chimeric spheroid, whereas those expression Eklin1-L210F were not. These data have been added as Figure 8—figure supplement 1.

4) The authors seem to suggest that Elkin1 knockout elicits an epithelial-to-mesenchymal transition. However, as the authors point out, the WT melanoma cells which express Elkin1 already show some mesenchymal characteristics. Moreover, the KO cells show a propensity for dissociating from the spheroid (consistent with an EMT phenotype), but also slower migration (contradictory to cancer cells that have undergone EMT). Given these ambiguities, it is better to eliminate the data on EMT markers (Figure 8A and subsection “Deletion of Elkin1 changes expression of epithelial and mesenchymal markers and modulates cell binding forces”) from the present study.

We have removed these details from the manuscript and additionally deleted the corresponding figure supplement with full blots and adjusted the Materials and methods to remove the corresponding experimental details. We thank the reviewers for the suggestion as we agree that these data introduced ambiguities to the manuscript.

5) Hoechst labeling of live cell experiments may compromise cell health, especially over long time-lapse imaging. The authors should include details on live-cell imaging (e.g. interval during images and duration of illumination for each time point) and ideally also confirm the results with cells labeled with a different fluorophore that is gentler on cells, e.g. GFP labeled cells.

The original design of this experiment was informed by published data from other groups where Hoechst was used as a marker for live cell studies (e.g. Harris and Nelson, 2010; Wolf et al., 2013). However, in order to address this point from reviewers we have repeated the 2D migration experiments using the virally-transduced clones constitutively expressing cytoplasmic GFP (see Materials and methods section). We found that differences in migration between the WT and KO clones was consistent with the original data set, i.e. the Elkin1-KO clone exhibited a slower track mean speed than the WT clone (as indicated in the figure supplement file that has now been added- see Figure 6—figure supplement 1). The illumination times have now been added to the Materials and methods section, the interval between images was described in the original submission and remains in the Materials and methods section. We would like to thank the reviewers for this suggestion as we did note that the GFP labelled cells exhibited a faster track mean speed than the corresponding cells labelled with Hoechst.

6) The authors conclude that knockdown of Elkin1 in WM266-4 cells results in reduction in MA currents, however in Figure 2A the final data point on the graph is nearly identical between WT and cells treated with Elkin1 miRNA. This supports the idea that Elkin1 may be a modulator of a MA channel and knockdown in Elkin1 expression results in a rightward shift in the sensitivity of the MA currents. The authors should address this possibility and state more clearly within the discussion that the role of Elkin1 in this mechanotransduction pathway is still not known.

We would like to first draw the reviewers’ attention to the difference between the knockdown (now Figure 2B, originally Figure 2A) and the knockout (in Figure 5B). When the expression of Elkin1 is completely abrogated by the CRISPR/Cas9 deletion (no mRNA detected) the final data point is significantly smaller in the knockout versus the wild type (Figure 5B). In contrast, as the reviewers note, in the knockdown the final point is similar to the controls (Figure 2B). We contend that this observation is due to an incomplete knockdown of *Elkin1*, as demonstrated by the qPCR experiments that we have now shifted to Figure 2A (previously in Figure 1—figure supplement 2).

To address the potential role of Elkin1 in this mechanotransduction pathway we used the following descriptions:

- In the Results section we interpreted the impact of Elkin1 knockdown as “[suggesting] that Elkin1 *contributes* to MA currents in melanoma cells.”

- In the section describing the impact of CRISPR/Cas9 mediated deletion of Elkin1 we wrote: “A residual current at large deflections was still present in some KO cells; we hypothesise that this effect is due to minor compensation from PIEZO1 (Supplementary file 1 and as noted previously (Servin-Vences et al., 2017)) or the activity of an as-yet-unidentified MA channel.”

- We additionally stated in the Discussion section “Elkin1 either mediates these MA currents or modulates an unidentified MA channel.”

In response to this concern, however, we have edited the Discussion to reiterate that Elkin1 may be modulating the function of an as yet unidentified MA channel, with the following text: “However, it is possible that Elkin1 is an accessory molecule that modulates the activation of an, as yet uncharacterised, MA channel or an ion channel that requires additional proteinaceous tethers in order to be activated by mechanical inputs. Future studies with purified protein will be required to definitively test the hypothesis that Elkin1 is an ion channel activated by mechanical inputs.”

We feel that we have clearly indicated, at each relevant point of the manuscript, that while Elkin1 may be an MA ion channel it may also be acting as a channel regulator and hope that this adjustment to the Discussion addresses the reviewers’ concern.